# Galectin-8 modulates human osteoclast activity partly through isoform-specific interactions

Michèle Roy[1] , Léopold Mbous Nguimbus[1] , Papa Yaya Badiane[1], Victor Goguen-Couture[1], Jade Degrandmaison[1], Jean-Luc Parent[1], Marie A Brunet[2], Sophie Roux[1]

In overactive human osteoclasts, we previously identified an alternative splicing event in *LGALS8*, encoding galectin-8, resulting in decreased expression of the long isoform. Galectin-8, which modulates cell–matrix interactions and functions intracellularly as a danger recognition receptor, has never been associated with osteoclast biology. In human osteoclasts, inhibition of galectin-8 expression revealed its roles in bone resorption, osteoclast nuclearity, and mTORC1 signaling regulation. Galectin-8 isoform-specific inhibition asserted a predominant role for the short isoform in bone resorption. Moreover, a liquid chromatography with tandem mass spectrometry (LC-MS/MS) proteomic analysis of galectin-8 isoforms performed in HEK293T cells identified 22 proteins shared by both isoforms. Meanwhile, nine interacting partners were specific for the short isoform, and none were unique to the long isoform. Interactors specific for the galectin-8 short isoform included cell adhesion proteins and lysosomal proteins. We confirmed the interactions of galectin-8 with CLCN3, CLCN7, LAMP1, and LAMP2, all known to localize to secretory vesicles, in human osteoclasts. Altogether, our study reveals direct roles of galectin-8 in osteoclast activity, mostly attributable to the short isoform.

## Introduction

Osteoclasts, large multinucleated cells (MNCs) formed by the fusion of precursors of the monocyte-macrophage lineage, are the only cells responsible for bone resorption (Boyle et al, 2003). In Paget's disease of bone (PDB) characterized by excessive and chaotic bone remodeling, osteoclasts are highly resorptive, and their phenotype includes PDK1/Akt–dependent mTORC1 activation, apoptosis resistance, and defective autophagy (McManus et al, 2016). In a previous study, we identified six alternative splicing (AS) events associated with the pagetic phenotype in PBMC-derived osteoclasts. Among them, the AS of *LGALS8*, which encodes for galectin-8,

was associated with decreased expression of the long isoform, even though total galectin-8 expression was increased in PDB osteoclasts compared with control osteoclasts (Klinck et al, 2014).

Although several galectins repress osteoclastogenesis and osteoclast function, including galectin-1 and galectin-9 (Moriyama et al, 2014; Muller et al, 2019), most osteoclast data are related to galectin-3 in mice. Galectin-3–deficient mice exhibit increased bone resorption, consistent with the suppressive effect of recombinant galectin-3 on osteoclastogenesis (Li et al, 2009; Simon et al, 2017). The expression and role of galectin-8 in osteoclasts remain unknown, although it was demonstrated to promote bone resorption in mice indirectly through the production of RANKL by galectin-8–stimulated osteoblasts (Vinik et al, 2015, 2018). Like all galectins, galectin-8 binds to $\beta$-galactoside and interacts with membrane glycoproteins and glycolipids but in addition binds to cytosolic and nuclear noncarbohydrate ligands (Johannes et al, 2018). The best characterized functions of galectin-8 are performed extracellularly, where it binds glycans to modulate cell–matrix interactions and trigger $\beta$1 integrin–mediated signaling cascades and cytoskeletal organization (Elola et al, 2014; Johannes et al, 2018). Galectin-8 has been implicated in cell spreading (Carcamo et al, 2006; Levy et al, 2006) and in the modulation of both innate and adaptive immune responses (Tribulatti et al, 2020) and neutrophil functions (Nishi et al, 2003). In addition to its involvement in the activation of dendritic cells and antigen-specific costimulation (Carabelli et al, 2017, 2018), galectin-8 also appears to be a potent proapoptotic agent in T cells and inflammatory cells (Eshkar Sebban et al, 2007; Norambuena et al, 2009). The functions of intracellular galectin-8 are not well known; however, cytosolic galectin-8 acts as a danger recognition receptor. Although the cytosol is devoid of complex carbohydrates under physiological conditions (Helenius & Aebi, 2001), damaged cytoplasmic vesicles, regardless of whether they are related to pathogens, expose sugars that recruit galectin-8, which appears to be a critical component for activating selective autophagy (Huang & Brumell, 2012; Thurston et al, 2012). By recruiting the adaptor NDP52, bound galectin-8 targets damaged vesicles for intracellular trafficking to autophagosomes in a ubiquitin-independent manner (Thurston et al, 2012). Galectin-8

[1]Division of Rheumatology, Department of Medicine, Faculty of Medicine and Health Sciences, University of Sherbrooke, Sherbrooke, Canada [2]Department of Paediatrics, Faculty of Medicine and Health Sciences, University of Sherbrooke, Sherbrooke, Canada

Correspondence: Sophie.Roux@USherbrooke.ca

also acts as an escort protein for K-Ras and participates in the regulation of its downstream signaling, enabling K-Ras to localize to vesicles and promote ERK 1/2 signaling (Meinohl et al, 2019). Galectin-8 could also be involved in the trafficking of proteins other than GTPases as it enables the apical localization of podocalyxin in renal epithelial cells (Lim et al, 2017).

*LGALS8* AS leads to the formation of distinct splice variants encoding two validated isoforms that share identical carbohydrate recognition domains (CRDs, N-terminal and C-terminal) and differ only in their linker region (insertion of 9 and 42 amino acids in the linker peptide of mouse and human galectin-8, respectively). The length and structure of the linker domain might alter the galectin-8/NDP52 interaction (Kim et al, 2013) and influence its biological function (Levy et al, 2006; Zhang et al, 2015; Si et al, 2023). Moreover, human galectin-8 with the longest linker peptide is highly susceptible to thrombin cleavage, whereas the other isoform is resistant (Nishi et al, 2006). Considering that *LGALS8* AS has been associated with a hyperactive osteoclast phenotype and is involved in pathways known to be critical in osteoclasts, such as autophagy and adhesion, our study aimed to assess the direct impact of galectin-8 and its isoforms on osteoclast phenotypes and identify interaction partners for each isoform to better understand the differential role of galectin-8 variants in these cells.

# Results

## Temporal expression and localization of galectin-8 in human osteoclasts

The presence of AS events in *LGALS8* led to the first evidence of galectin-8 expression by human osteoclasts (Klinck et al, 2014). In Fig 1A, a linear representation showcases the two *LGALS8* isoforms. Galectin-8 expression was initially evaluated at various stages of differentiation. However, for the remainder of the study, all experiments will exclusively target mature osteoclasts. Using a non-isoform-specific antibody, we therefore assessed galectin-8 and distinguished its long and short isoforms based on their molecular weights by immunoblotting at different time points during osteoclast long-term cultures (Fig 1B). Total galectin-8 expression increased over time, reaching a plateau after 14 d of culture for both isoforms, with the short form accounting for ~60% of all galectin-8. Although *LGALS8*-short isoform represented the predominant expression throughout the 21 d of culture, a significant change in the isoform ratio was observed from day 7 (Fig 1C and D). Moreover, by immunofluorescence analysis, we observed a dot-like cytoplasmic staining pattern of galectin-8 from the top (functional secretory domain) to the apical domain and within the sealing zone defined by F-actin, which delimits the ruffled membrane in active osteoclasts seeded on bone slices (Fig 1E).

## Inhibiting galectin-8 using RNA interference

To explore the effect of galectin-8 on the multinuclearity and bone resorption of late-differentiating osteoclasts (day 17), siRNAs were used to reduce its expression globally or in an isoform-specific manner. Transfection of Dicer-substrate siRNA (DsiRNA) sequences targeting human *LGALS8* resulted in a 75% decrease in its expression (Fig S1A and B). To specifically down-regulate each isoform, osteoclast cultures were then transfected with siRNAs targeting either the short or the long isoform (Fig S1C). When transfecting the long isoform–specific siRNA, expression of *LGALS8*-long was decreased by ~74% versus the control. Meanwhile, siRNA specific for the short isoform decreased its expression by an average of 64%, with resulting changes in the isoform ratio of 20–25% (Fig S1D and E).

## Inhibiting galectin-8 expression in osteoclasts decreases in vitro nuclearity and bone resorption

To assess the impact of galectin-8 on the phenotype of mature osteoclasts, we evaluated multinucleation and bone resorption—the exclusive specific marker for mature osteoclasts—in cultures where osteoclasts were transfected during the late stages of their maturation (day 17). By reducing the expression of galectin-8, we observed significant decreases in the number of MNCs (−40%) and the number of nuclei per MNC compared with the findings in control cultures (Fig 2A–C). MNCs were divided into three groups according to the number of nuclei: low (3–5), medium (6–10), and high (≥11). We found a significant decrease in the proportion of high-nucleated cells in the presence of galectin-8 DsiRNAs compared with the samples treated with a control DsiRNA, whereas the proportion of MNCs with medium and low nucleus numbers was increased (Fig 2D). Specific inhibition of the galectin-8 long isoform did not affect the number of MNCs, but targeting the short isoform induced a significant decrease in the number of MNCs (−22%) versus the control (Fig 2E and F). Although a slight decrease in the number of nuclei per MNC was observed in all transfected cells, this decrease was significant only when inhibiting the short isoform compared with the control (Fig 2G). No significant changes were observed in the proportions of low-, medium-, and high-nucleated cells across transfected cells, although a trend of fewer high-nucleated cells was observed with both siRNAs (Fig 2H).

When osteoclasts were cultured on bone slices, inhibition of galectin-8 expression resulted in significant reductions in the resorbed bone area (−78%) and resorption area per MNC (−63%) compared with negative control and untransfected osteoclasts (Fig 3A and B). When inhibiting either galectin-8 isoform, siRNA targeted at the long isoform did not produce a significant change in the resorbed area. In contrast, short isoform–specific siRNA significantly decreased the resorbed area (−39%) compared with control siRNA, resulting in a 26% decrease in the resorbed bone area per MNC (Fig 3C and D).

As a classical role of galectins is to promote adhesion, and to better understand how the resorption process could be affected by galectin 8, we investigated by immunofluorescence the sealing zone in osteoclasts cultured on the bone (Fig 4A), measuring both the surface area of this zone (Fig 4B) and the density of the actin-rich podosomes comprising it (Fig 4C). Reducing galectin-8 expression resulted in a disorganized sealing zone, showing a decrease in area and an increased distance between podosomes. Targeting the expression of either the short or long isoform led to a similar decrease in the distance between podosomes.

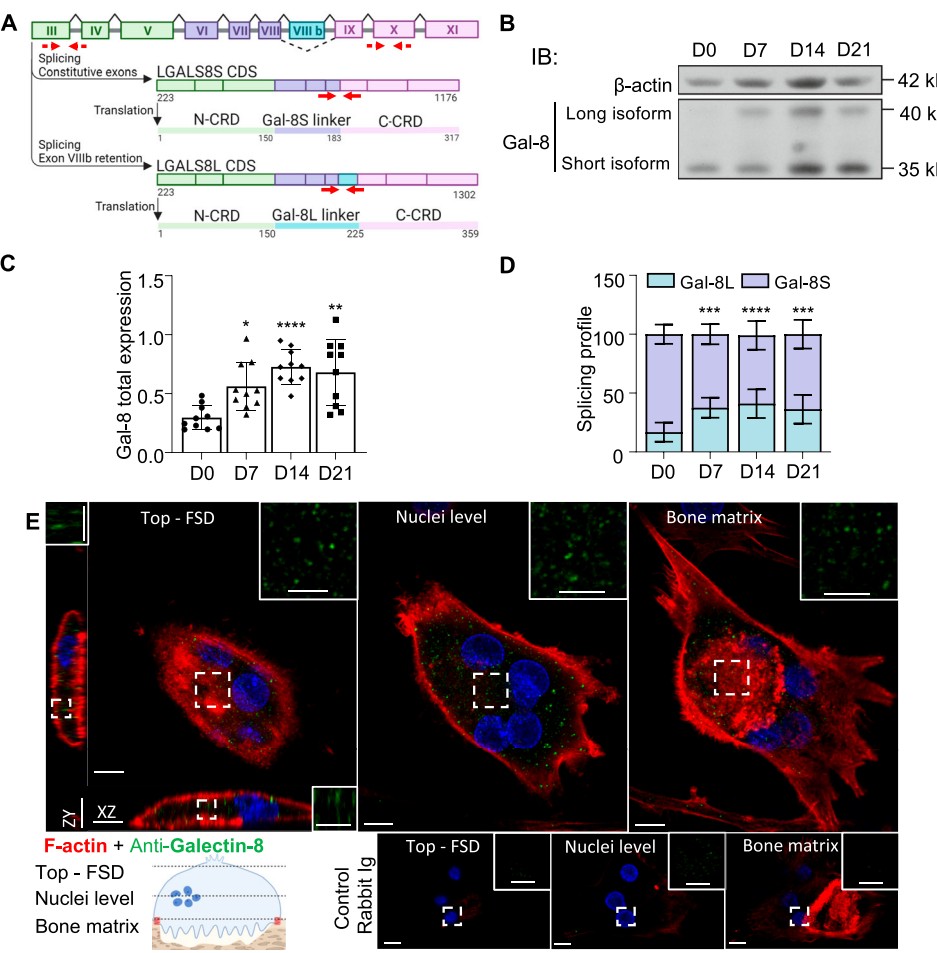

**Figure 1. Expression of galectin-8 and its isoforms in human osteoclasts.**
**(A)** Alternative splicing of the human *LGALS8* gene. Diagram showing the position of exon VIIIb (blue), spliced in the short *LGALS8S* isoform and retained in the long *LGALS8L* isoform (42 aa longer peptide linker). Coding sequence and primary structure of galectin-8 isoforms are presented. The red lines mark the RNA interference experiment targets: solid for isoform-specific siRNAs and dashed for DsiRNA. **(B)** Time-dependent expression of galectin-8 during osteoclast differentiation, analyzed by immunoblotting (IB) using antibodies recognizing both galectin-8 isoforms or β-actin. **(C)** Total galectin-8 (long + short) relative expression over β-actin is presented. **(D)** The splicing profile was determined using the protein ratio of the long isoform to total galectin-8. All results are expressed as the mean ± SD (N = 10 independent experiments; *P < 0.05, **P < 0.01, ***P < 0.001, ****P < 0.0001 versus D0); paired one-way ANOVA (C), two-way ANOVA (D). **(E)** Galectin-8 expression in osteoclasts cultured on bone slices, using double-staining immunofluorescence with phalloidin (red), antibodies directed against galectin-8 (green), negative control (rabbit Ig), and DAPI (blue). Optical Z-sections of the top (functional secretory domain), nuclear, and bone matrix levels of an osteoclast are presented with its orthogonal reconstructions (scale bar = 10 *µ*M) and magnification of galectin-8 staining (scale bar = 5 *µ*M). Images are representative of three independent experiments with 10 acquired images per experiment.

Thus, inhibition of galectin-8 expression decreased the number of MNCs and bone resorption activity of osteoclasts, which could be mainly related to the function of the short isoform. Meanwhile, neither nuclearity nor the influence of galectin-8 on the actin-rich podosome belt seemed to be specific to a particular isoform.

### Galectin-8 inhibition in osteoclasts induces autophagy through mTORC1 signaling

Some functions of intracellular galectin-8 are closely associated with autophagy and mTORC1 signaling (Jia et al, 2018), a crucial pathway in osteoclasts. Indeed, stimulation of autophagy blocks osteoclast differentiation and bone resorption (Hussein et al, 2012), and PDK1-dependent mTOR hyperactivation with a defect in autophagy has been reported in pagetic overactive osteoclasts (McManus et al, 2016). We therefore evaluated osteoclast autophagy in the presence or absence of galectin-8 inhibition. The different proteins related to mTORC1 that were studied have been represented on a diagram (Fig 5A).

The global impact of galectin-8 inhibition on autophagy was first evaluated according to the expression of LC3B-II, a LC3–phosphatidylethanolamine conjugate, which is correlated with increased levels of autophagic vesicles (Barth et al, 2010). The LC3B-II/LC3B-I ratio was higher in galectin-8 DsiRNA–transfected cells

than in control DsiRNA–transfected cells, indicating that inhibiting galectin-8 expression increased basal autophagy in osteoclasts (Fig 5B and C). AMPKα phosphorylation at T[172], the expression of which was increased upon galectin-8 down-regulation, positively regulates autophagy via the inhibition of mTORC1 activity. As an upstream regulatory kinase of Raptor that inhibits mTORC1 complex signaling (Gwinn et al, 2008), AMPKα activation might account for the increase in Raptor phosphorylation at Ser[792] (Fig 5D). mTORC1 activation was evaluated through the phosphorylation of its targets 4EBP1 at Thr[37]/Thr[46] and serine/threonine kinase p70 S6K at Thr[229] and the phosphorylation of ULK1 at Ser[757], which prevents ULK1 activation, and autophagy induction (Hara et al, 2002). ULK1, 4EBP1, and p70 S6 kinase phosphorylation was significantly lower in osteoclasts transfected with galectin-8 DsiRNAs than in control DsiRNA–transfected or –untransfected cells, suggesting a dampening of mTORC1 signaling to mediate autophagy activation (Fig 5E). Finally, the expression of activated/phosphorylated forms of Akt, ERK, and PDK1 was assessed as upstream stimulators of mTORC1. Phosphorylation of Akt at Thr[308] is performed by PDK1, and this protein is secondarily phosphorylated within its C-terminus at Ser[473] by mTORC1, M-CSF, or RANKL (Glantschnig et al, 2003; Sarbassov et al, 2005). Phosphorylation of PDK1 at Ser[241] and Akt at Ser[473] was not changed by galectin-8 inhibition, whereas that of ERK1/2 at Thr[202]/Tyr[204] was significantly lower (Fig 5F).

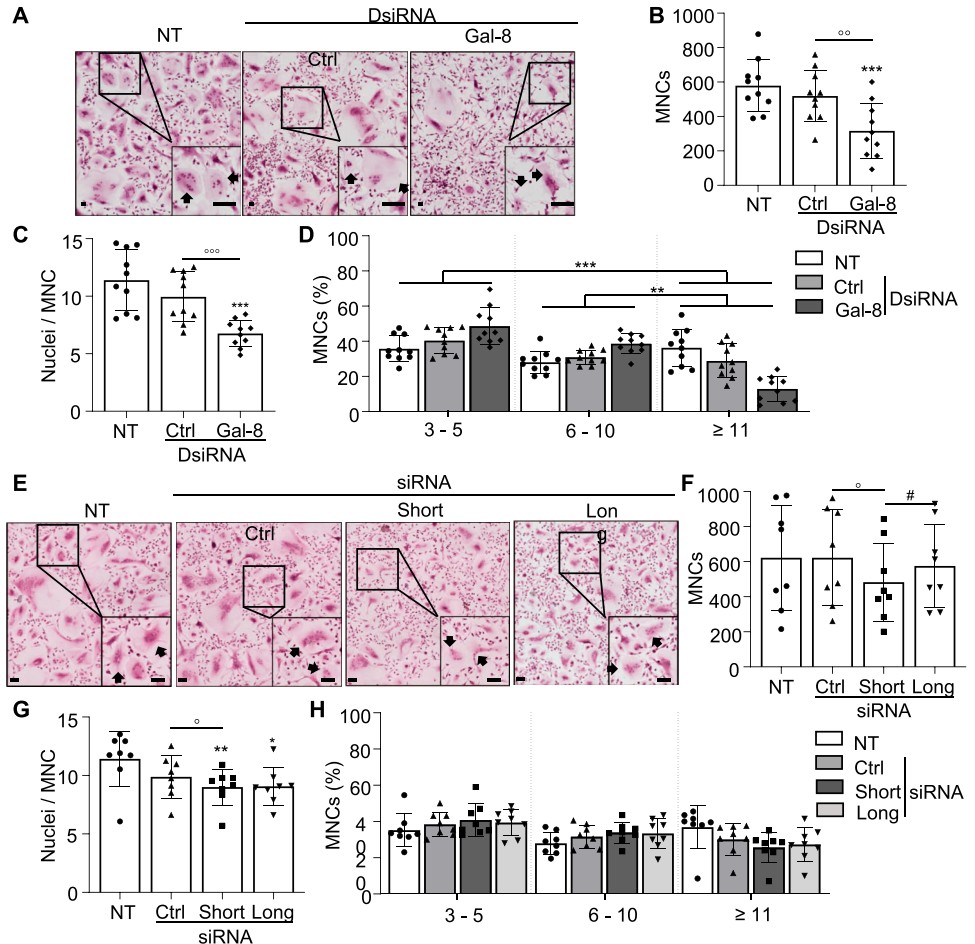

**Figure 2. Multinucleated cell (MNC) formation and galectin-8 inhibition.**
In osteoclast cultures transfected on day 17 with negative controls (Ctrl), total galectin-8 DsiRNAs or isoform-specific siRNAs or not transfected (NT), MNCs were analyzed on day 21. **(A)** Inhibition of galectin-8 global expression. Representative images of MNCs (arrows) are shown (scale bar: 0.5 mm). **(B)** Results are expressed graphically as the number of MNCs per well (surface area: 0.4 cm$^2$). **(C)** The results are graphically expressed as nuclei per MNC. **(D)** Percentages of MNCs with 3–5, 6–10, and ≥11 nuclei are presented on a graph representing the contingency table. **(E)** Effect of galectin-8 isoform–specific inhibition. Representative images of MNCs (arrows) are shown (scale bar: 0.5 mm). **(F)** Results are expressed graphically as the number of MNCs per well (surface area: 0.4 cm$^2$). **(G)** The number of nuclei per MNC (nuclei/MNC) is presented. **(H)** The graph presents the percentages of MNCs with 3–5, 6–10, or ≥11 nuclei. All results are expressed as the mean ± SD (N = 8–10 independent experiments, two replicates per experimental condition. $*P < 0.05$, $**P < 0.01$, $***P < 0.001$ versus NT; $^{o}p < 0.05$, $^{oo}p < 0.01$, $^{ooo}p < 0.001$ versus Ctrl; $\#P < 0.05$ short versus long); paired one-way ANOVA and chi-squared test (D, H).

Specific inhibition of each galectin-8 isoform did not affect autophagy as assessed globally by the LC3B-II/LC3B-I ratio nor did it affect the aforementioned mTORC1-related regulators and targets (Fig S2), indicating that autophagy and its associated signaling are not dependent on spliced galectin-8 isoforms, with the ~40% decrease in total galectin-8 expression not being sufficient to induce any change.

## Decreasing galectin-8 expression stimulates autophagy without altering autophagic flux

The high basal LC3B-II/LC3B-I ratio induced by galectin-8 inhibition could result from an increase in autophagy induction or defective autophagosome processing as LC3B-II is subject to autophagic degradation in lysosomes. This was investigated by measuring autophagic flux in galectin-8 DsiRNA–transfected and –untransfected osteoclast cultures, maintained in standard medium or treated with rapamycin or starvation to induce autophagy in the presence or absence of chloroquine (Fig 6A). First, the expression profile and splicing profile of galectin-8 in untransfected cells remained unchanged under all autophagic flux conditions, indicating that neither galectin-8 isoform is a substrate for autophagy (Fig 6B and C). The LC3B-II/LC3B-I ratio was significantly increased when

autophagy was induced by rapamycin and serum-free EMEM in untransfected cells and in galectin-8 or control DsiRNA–transfected cells compared with the findings in cells grown in standard medium. Inhibition of galectin-8 increased the basal autophagy rate, as indicated by a higher LC3B-II/LC3B-I ratio compared with that in control DsiRNA–transfected and -untransfected cells, with a further increase achieved in the presence of chloroquine. These results indicate that high LC3B-II/LC3B-I ratios induced by galectin-8 depletion were related to autophagy induction, as suggested by the previously observed reduction in mTORC1 signaling, without impairment of autophagy clearance (Fig 6D).

Finally, we studied ULK1 phosphorylation at Ser$^{757}$ upon autophagy induction with rapamycin or serum-free EMEM. A strong decrease in ULK1 phosphorylation at Ser$^{757}$ was observed when autophagy was induced by rapamycin or serum-free EMEM in all cell cultures compared with the findings in standard medium, with lower levels in observed cells transfected with galectin-8 DsiRNA than in control DsiRNA–transfected or –untransfected cells (Fig 6E).

## Differential interactome of galectin-8 isoforms

Overall, the reduced proportion of the long galectin-8 isoform in overactive pagetic osteoclasts (Klinck et al, 2014) and the

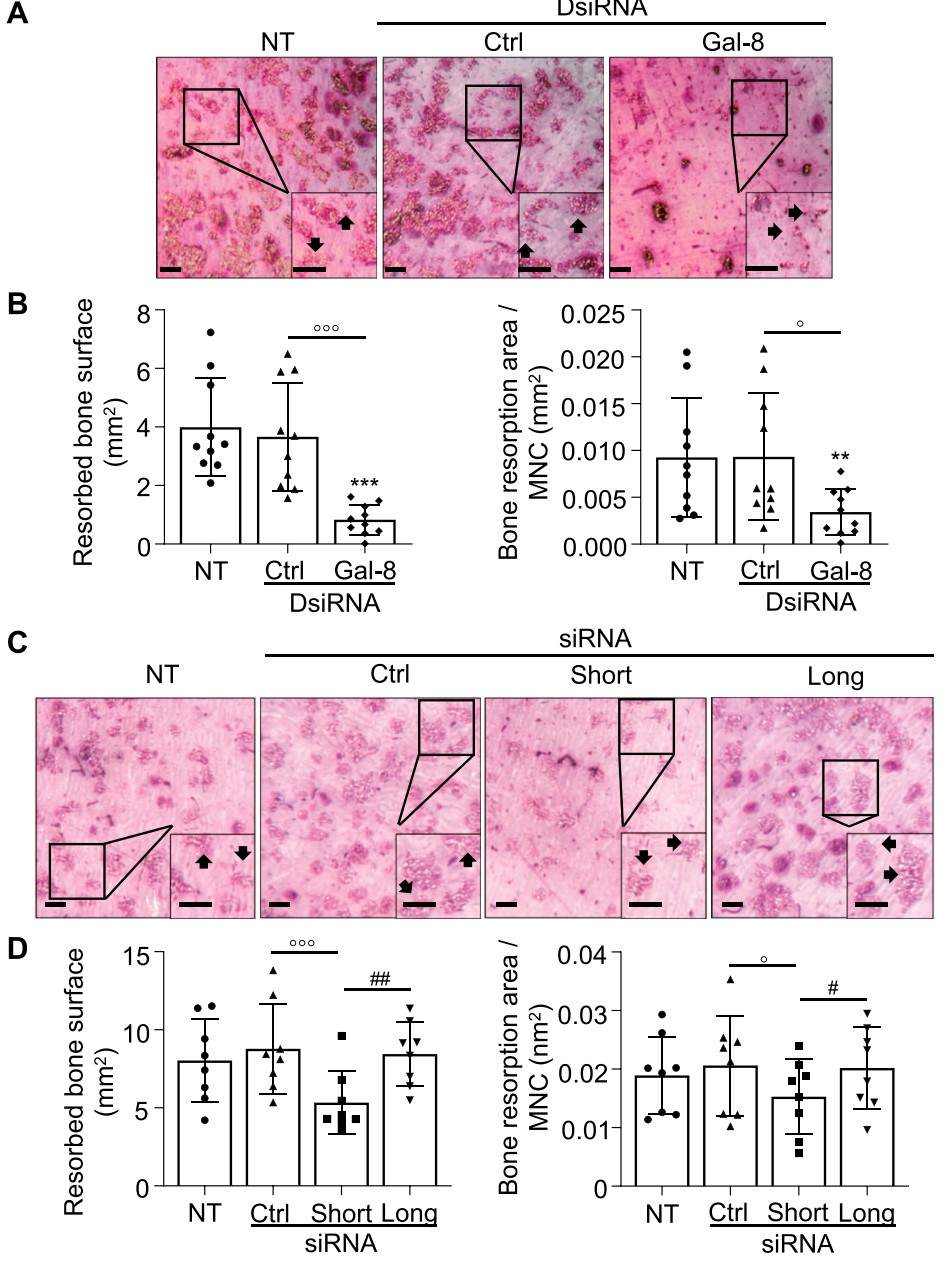

**Figure 3. Osteoclast bone resorption and galectin-8 inhibition.**
In osteoclast cultures performed on bone slices, and transfected on day 17 with negative controls (Ctrl), galectin-8 DsiRNA or isoform-specific siRNAs or not transfected (NT), bone resorption was analyzed on day 21. **(A)** Inhibition of galectin-8 global expression. Resorption areas appear blue/purple in color, and the brightness obtained in epi-illumination reflects the depth of bone resorption. Representative images illustrating bone resorption pits (arrows) are shown. **(B)** Resorbed bone area (mm$^2$) and bone resorption per multinucleated cell were measured. **(C)** Effect of galectin-8 isoform–specific inhibition: representative images illustrating bone resorption. **(D)** Resorbed bone area (mm$^2$) and bone resorption per multinucleated cell. All results are expressed as the mean ± SD (N = 8–10 independent experiments, two bone slices per experimental condition; $^o$p < 0.05, $^{ooo}$p < 0.001 versus Ctrl: $^#P$ < 0.05, $^{##}P$ < 0.01 short isoform siRNA versus long isoform siRNA); paired one-way ANOVA (B, D); scale bars: 0.5 mm.

preferential impact of the short galectin-8 isoform in vitro support the differential roles of galectin-8 isoforms in bone resorption and in the number of MNCs, whereas autophagy did not appear to be isoform-dependent in osteoclasts. The two isoforms might interact with distinct proteins and metabolic pathways, as indicated by the impact of linker length on galectin-8 functions (Levy et al, 2006), which led us to analyze the interactome of galectin-8 isoforms by liquid chromatography with tandem mass spectrometry (LC-MS/MS). As the anti-galectin-8 antibodies currently available do not allow isoform differentiation, and mature human osteoclasts in primary culture are excessively challenging to transfect (Laitala-Leinonen, 2005), to study the galectin-8 interactome of each isoform, we chose to use the model of HEK293T cells transfected with a plasmid containing cDNA of each isoform (Fig 7A).

### Identification of high-confidence interaction proteins (HCIP) specific for the galectin-8 short isoform

From the raw LC-MS/MS data, we first excluded interactions between the short and long isoforms by confirming the absence of unique peptides of each isoform in samples in which the other isoform was the bait. The results were then scored using SAINTexpress, with a score of at least 0.9 validating an interaction with one isoform and a score greater than 0.7 with the other isoform qualifying common interacting partners. Spearman's rank correlation matrices of biological replicates are also provided (Fig S3).

**A**

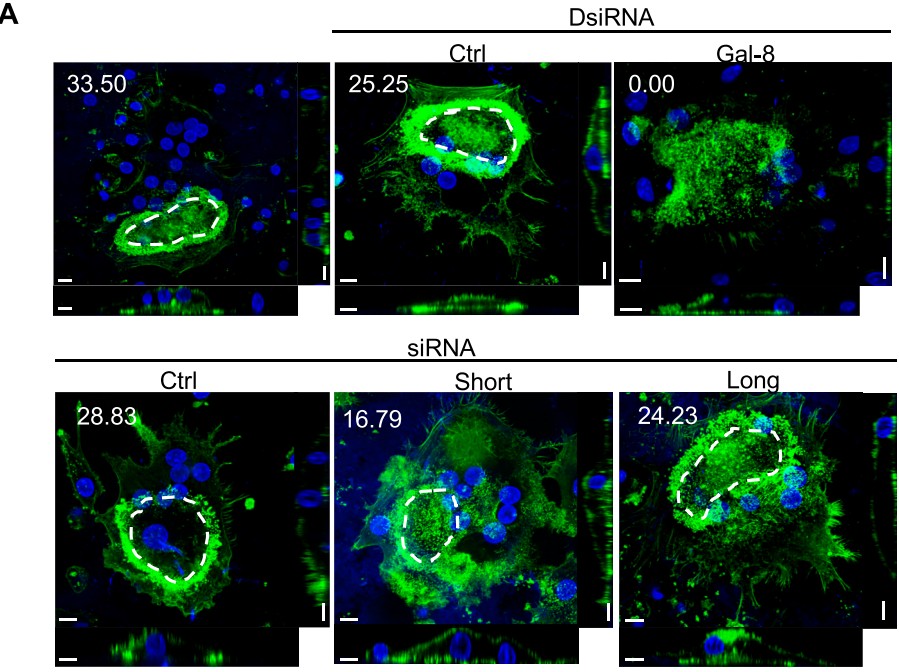

Figure 4. Impact of galectin-8 inhibition on sealing zone integrity.
(A) Osteoclast cultures, either transfected with negative controls (Ctrl), total galectin-8 DsiRNAs, isoform-specific siRNAs, or left non-transfected (NT) on day 17 were subjected to incubation with phalloidin Alexa Fluor 488 (green) and DAPI. The dashed lines indicate the region measured to quantify the actin ring area (scale bar = 10 $\mu M$).
(B) Quantification of the surface area outlined by the actin ring ($\mu m^2$).
(C) Quantification of the average distance between podosomes in a 10-$\mu M$ horizontal section of the actin ring ($\mu m$). Representative images are derived from three independent experiments, each comprising 30 acquired images per experimental condition. $^{o}p < 0.05$, $^{ooo}p < 0.001$, $^{oooo}p < 0.0001$ versus Ctrl, ****$P < 0.0001$ versus NT; paired one-way ANOVA (B, C).

**B** **C**

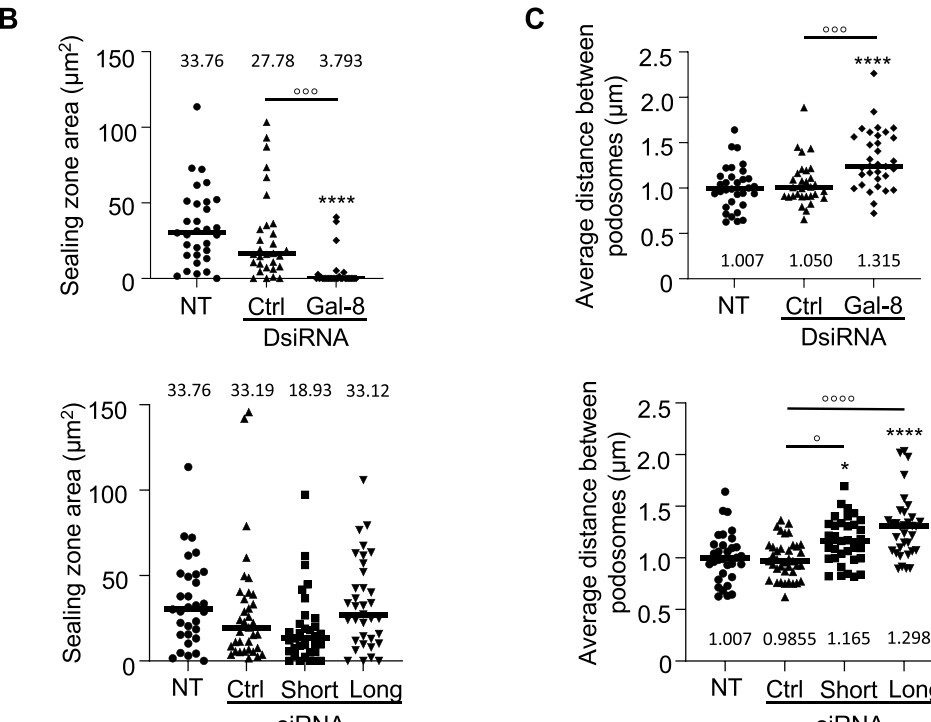

According to the selection criteria, 31 interacting proteins were identified, including 22 proteins shared by both isoforms and nine short isoform–specific proteins (Fig 7B). The common interacting proteins included classical partners of galectin-8, such as integrin beta 1 (ITGB1), proteins involved in autophagy (ATG9A, LAMP2), and numerous membrane transporters. Specific proteins for the galectin-8 short isoform included integrins and other proteins involved in cell adhesion (ITGA4, ITGA5, TPBG), lysosomal proteins (LAMP1, SCL17A5), and membrane receptors (M6PR, PTGFRN) (Table S1).

To better understand the differences in binding partners between the isoforms, we first assessed the N-CRD/C-CRD ratios (peptide-spectrum match of single peptides of N-CRD versus C-CRD domain

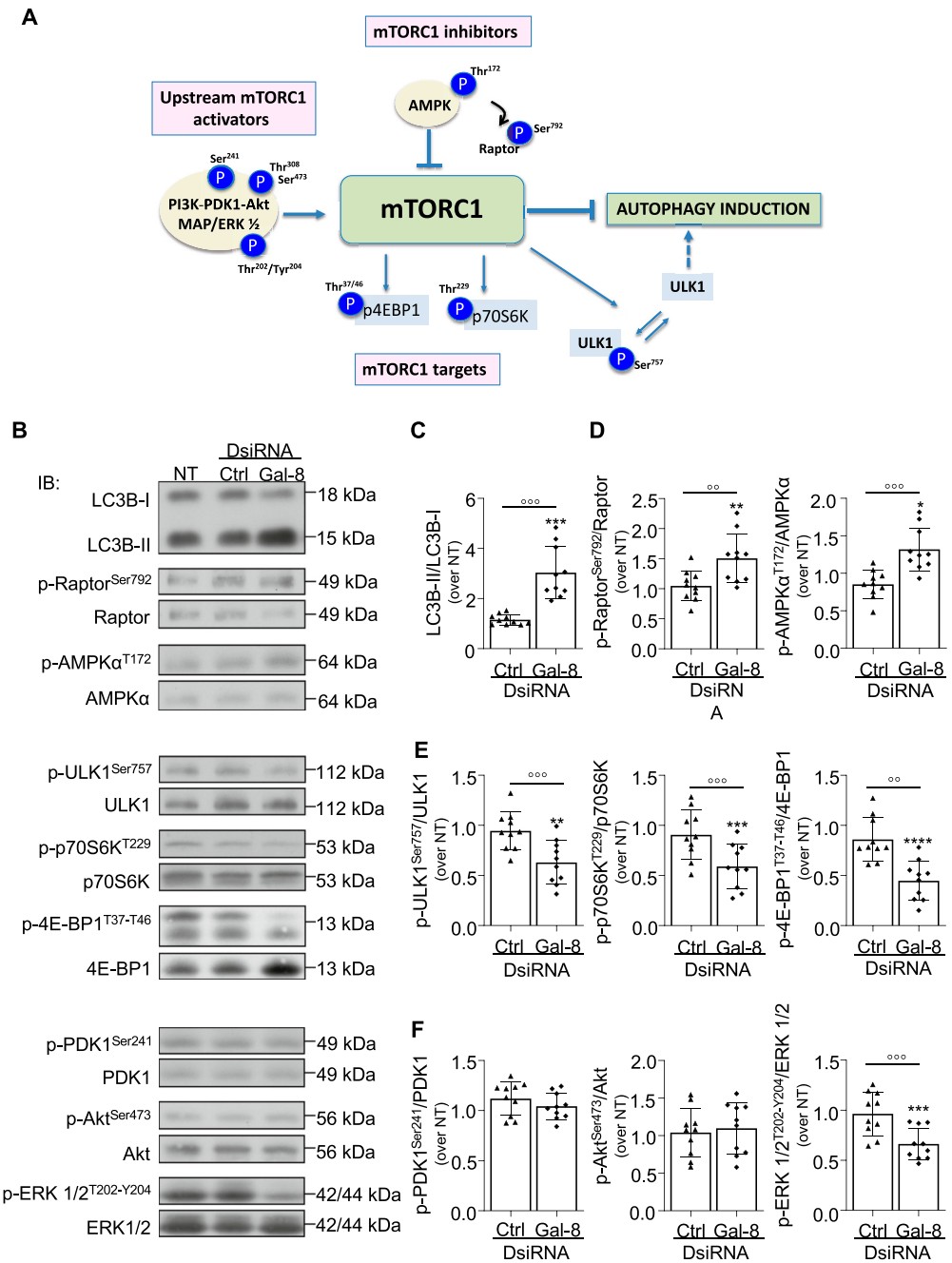

**Figure 5. Impact of galectin-8 inhibition on mTORC1 signaling.**
On day17, osteoclast cultures were transfected with galectin-8 DsiRNAs (Gal-8) or negative control DsiRNA (Ctrl) or not transfected (NT), and mTORC1 signaling and LC3B were evaluated by immunoblotting and the ratios of the phosphorylated (p) over total forms of several key proteins are presented. **(A)** Diagram. The organizational chart of the mTORC1-related proteins/p-proteins studied is schematized. **(B)** Analysis of autophagy- and mTOR-related proteins by immunoblotting. **(C)** Autophagy levels. The graph shows the ratio of LC3B-II to LC3B-I as an indicator of basal autophagy. **(D)** mTORC1 inhibitors (AMPKα and its target Raptor). **(E)** mTORC1 targets. The graphs show the phosphorylation of downstream substrates of mTORC1, including ULK1, p70S6K, and 4E-BP1. **(F)** Upstream mTORC1 activators. The graphs show the phosphorylation of kinases upstream of the mTORC1 complex, including PDK1, Akt, and ERK. Results are expressed as the mean ± SD (N = 10 independent experiments; *$P < 0.05$, **$P < 0.01$, ***$P < 0.001$, ****$P < 0.0001$ versus NT; °°$p < 0.01$, °°°$p < 0.001$ versus Ctrl); paired one-way ANOVA (C, D, E, F).

in biological replicates), which were not significantly different, indicating that both CRDs were equally represented in transfected HEK293T cells (Table S2). We then investigated whether the tertiary structure could explain such a difference in interacting proteins. The 3D structure of each isoform was therefore generated by AlphaFold2 using the local predicted distance difference test to measure the confidence of the models by residue (Tunyasuvunakool et al, 2021). The loop presented for the long isoform linker peptide reflected the lack of a clear configuration, which remains unknown because of its high degree of flexibility as previously reported (Gomez-Redondo et al, 2021; Si et al, 2023) (Figs 7C and S4).

### GO (Gene Ontology) and KEGG (Kyoto Encyclopedia of Genes and Genomes) pathway enrichment analyses

Finally, the functional differences between the long and short isoforms of galectin-8 were studied through enrichment analysis of GO terms and metabolic pathways, including all interactors (common and specific) for the short isoform and those only shared by the long isoform (Fig 7D). To simplify the reading of the data representation, some GO terms were grouped according to their similarity (Table S3). GO enrichment analyses showed that galectin-8 isoforms shared several common functions, some of which were well characterized or relevant in osteoclasts (Ribet et al, 2021), such

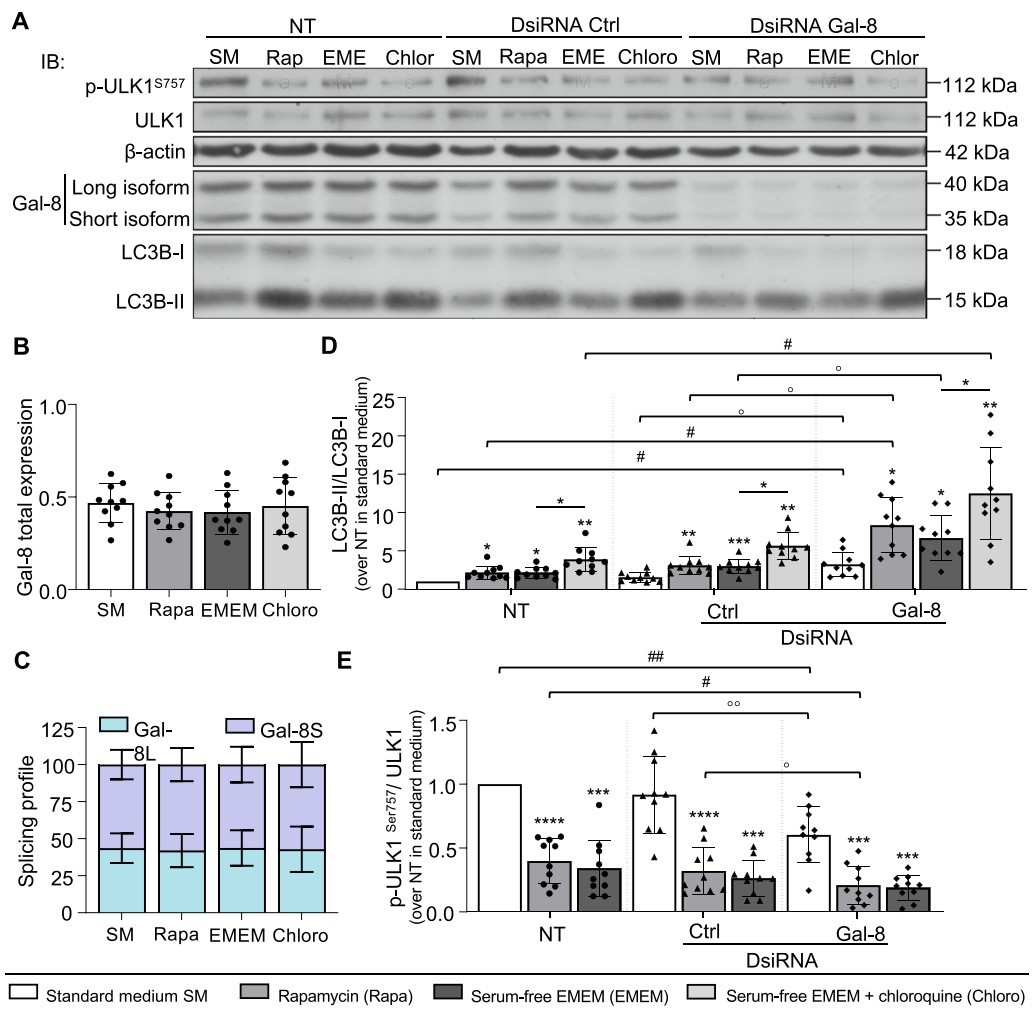

**Figure 6. Effect of inhibiting galectin-8 expression on autophagy flux.**
Osteoclast cultures were transfected with either galectin-8 DsiRNAs (Gal-8) or negative control DsiRNA (Ctrl) or not transfected (NT) on day 17. On day 21, cells were cultured in standard medium (SM) and exposed to either rapamycin (Rapa) or serum-free EMEM (EMEM) to induce autophagy, with or without chloroquine (Chloro). **(A)** Impact of galectin-8 on autophagy-related protein expression. ULK1 and its phosphorylated (Ser[757]) form, galectin-8, LC3B, and β-actin were analyzed by immunoblotting. **(B)** The graphs show the ratio of total galectin-8 (long + short) protein expression to β-actin. **(C)** Galectin-8 splicing profile determined as the ratio of the long isoform to total galectin-8. **(D)** LC3B-II/LC3B-I ratio. **(E)** Ser[757]p-ULK1/ULK1 ratio. Data are presented as the mean ± SD (N = 10 independent experiments; $*P < 0.05$, $**P < 0.01$, $***P < 0.001$, $****P < 0.0001$ versus SM of the same transfection condition; $^{\#}P < 0.05$, $^{\#\#}P < 0.01$ versus NT, same autophagy stimulation; $^{o}p < 0.05$ Ctrl versus Gal-8 DsiRNA); paired one-way (B, D, E) or two-way (C) ANOVA.

as cell adhesion, membrane trafficking, and phagosome and lysosome metabolic pathways.

### Co-localization of short isoform–associated proteins in osteoclasts

Chosen proteins, like CLCN3, LAMP2 (shared), CLCN7, and LAMP1 (specific to the short isoform), were selected to validate the human osteoclast interactome based on data initially obtained from HEK293T cells. These proteins were selected for further study considering the highly represented GO terms associated with lysosomal proteins and membrane trafficking and associated pathways of the interactomes. Moreover, the chloride channels CLCN3 and CLCN7 are known to participate in bone resorption because CLCN7 ablation leads to osteopetrosis in humans (Kornak

et al, 2001) and CLCN3 contributes to organelle acidification in mice (Okamoto et al, 2008). In addition, the membrane proteins LAMP1 and LAMP2 are expressed in lysosomes and autophagosomes, and they were localized to secretory vesicles associated to bone resorption along with CLCN3/CLCN7, at least in mice (Ng et al, 2023).

Importantly, LAMP1 was highly specific for the short isoform (SAINT score: 0.99 for the short isoform versus 0 for the long isoform), whereas LAMP2 and CLCN3 were among the strongest galectin-8 interactors (SAINT score of 1.00 for both isoforms). Although the SAINT score for CLCN7 was below our stringent threshold, it was selected because this interactor is specific for the short isoform (SAINT score: of 0.74 for the short isoform versus 0 for the long isoform) and because of its crucial role in bone resorption (Table S1).

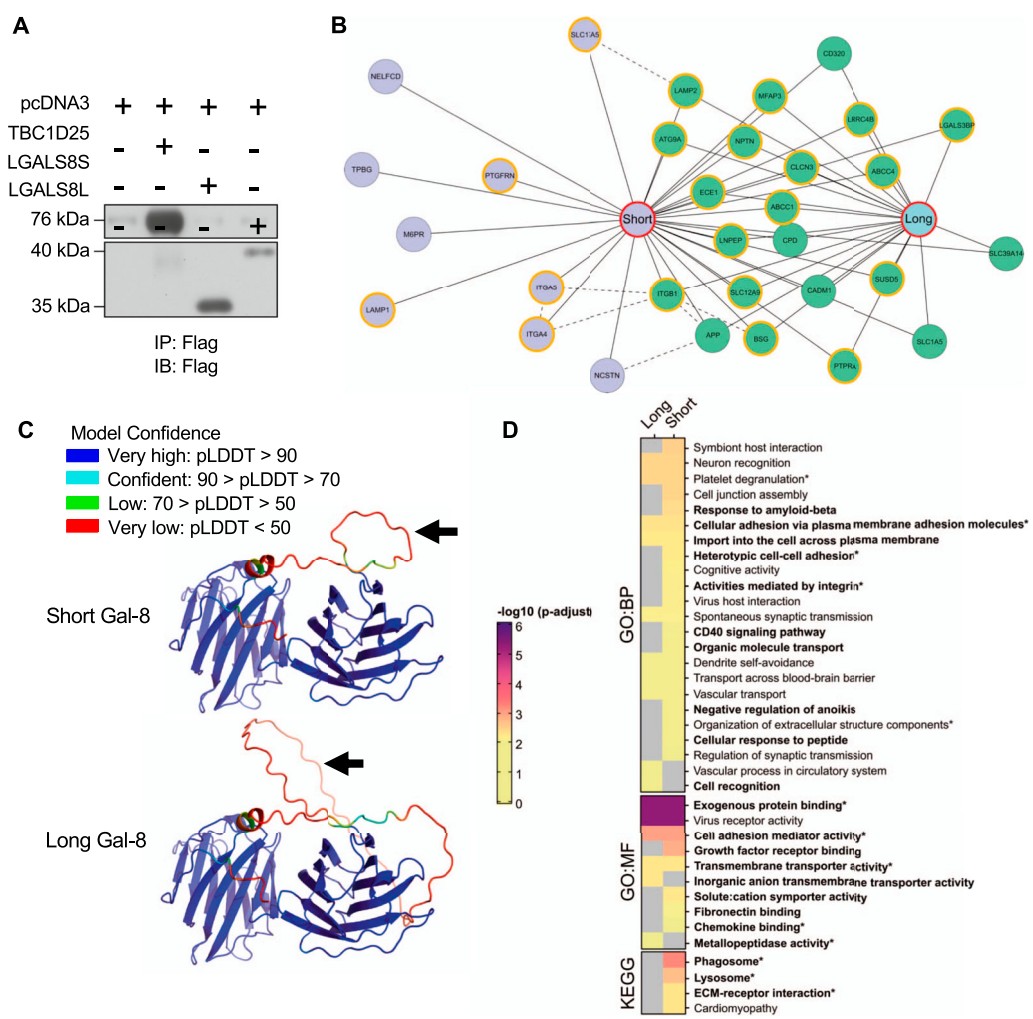

**Figure 7. Identification of proteins interacting with galectin-8 isoforms.**
**(A)** Affinity purification–mass spectrometry: sample process. HEK293T cells were transfected with pcDNA3, FLAG-*TBC1D25* (*TBC1D25*), FLAG-*LGALS8*-short (*LGALS8S*), or FLAG-*LGALS8*-long (*LGALS8L*). Immunoprecipitation (IP) and immunoblotting (IB) were analyzed using an antibody against the FLAG epitope. **(B)** Interactomes of the short and long galectin-8 isoforms. Interacting proteins are presented as circles (green if common, blue if short isoform–specific, orange outline if previously reported), and each isoform is represented by a red circle. The dashed lines represent prey interactions according to the STRING database (combined score > 0.75). **(C)** Predicted 3D isoform structures. The 3D predictive structures of the short and long isoforms of galectin-8 according to their linker sequence (arrows) were generated via AlphaFold2 using local predicted distance difference test. **(D)** GO enrichment analyses of common and specific interacting proteins of the short and long isoforms of galectin-8. The heatmap displays all significant enrichments, and nonenriched GO terms are presented in gray. GO terms presented in boldface are relevant to osteoclast biology, whereas asterisks indicate previously reported functions of galectin-8. BP, biological processes; MF, molecular functions.

As mentioned previously, galectin-8 displayed a dot-like cytoplasmic staining pattern from the top (functional secretory domain) to the apical domain in active osteoclasts (Fig 1F). CLCN7 was highly expressed near the bone matrix at the ruffled membrane surrounded by F-actin structures, which characterize the sealing zone (Fig 8A). Conversely, CLCN3 was strongly expressed at the nuclear level and found to be diffused in the cytoplasm between the nuclear level and the apical domain (Fig 8B). To assess the interaction between galectin-8 and its interactors, Mander's co-localization coefficient (R) was obtained from the nuclear level, bone matrix level, and orthogonal reconstruction. Because we observed nonspecific staining in nuclei, they were excluded from the region of interest when assessing co-localization. Galectin-8 co-localized with CLCN3 at different levels in the cytoplasm between nuclei

and the bone matrix (R = 0.35–0.54) (Fig S5A and D). The co-localization between galectin-8 and CLCN7 was mostly restricted to the ruffled membrane (R = 0.72) (Fig S5B and D). By interacting with CLCN7 at the ruffled membrane, the short isoform might play a direct and greater role in bone resorption.

Furthermore, we investigated the interactions of galectin-8 with LAMP1 and LAMP2. Surprisingly, in all experiments, although detection was clear in mononuclear cells, LAMP1 was detected only in a few osteoclasts, in which it localized near galectin-8 at the periphery of the ruffled membrane (Figs 9A and S6). As expected, LAMP2 was strongly detected at the ruffled membrane with faint expression at the nuclear level (Fig 9B). Galectin-8 co-localized to some extent with LAMP2 at the ruffled membrane, as observed in sections acquired near the bone

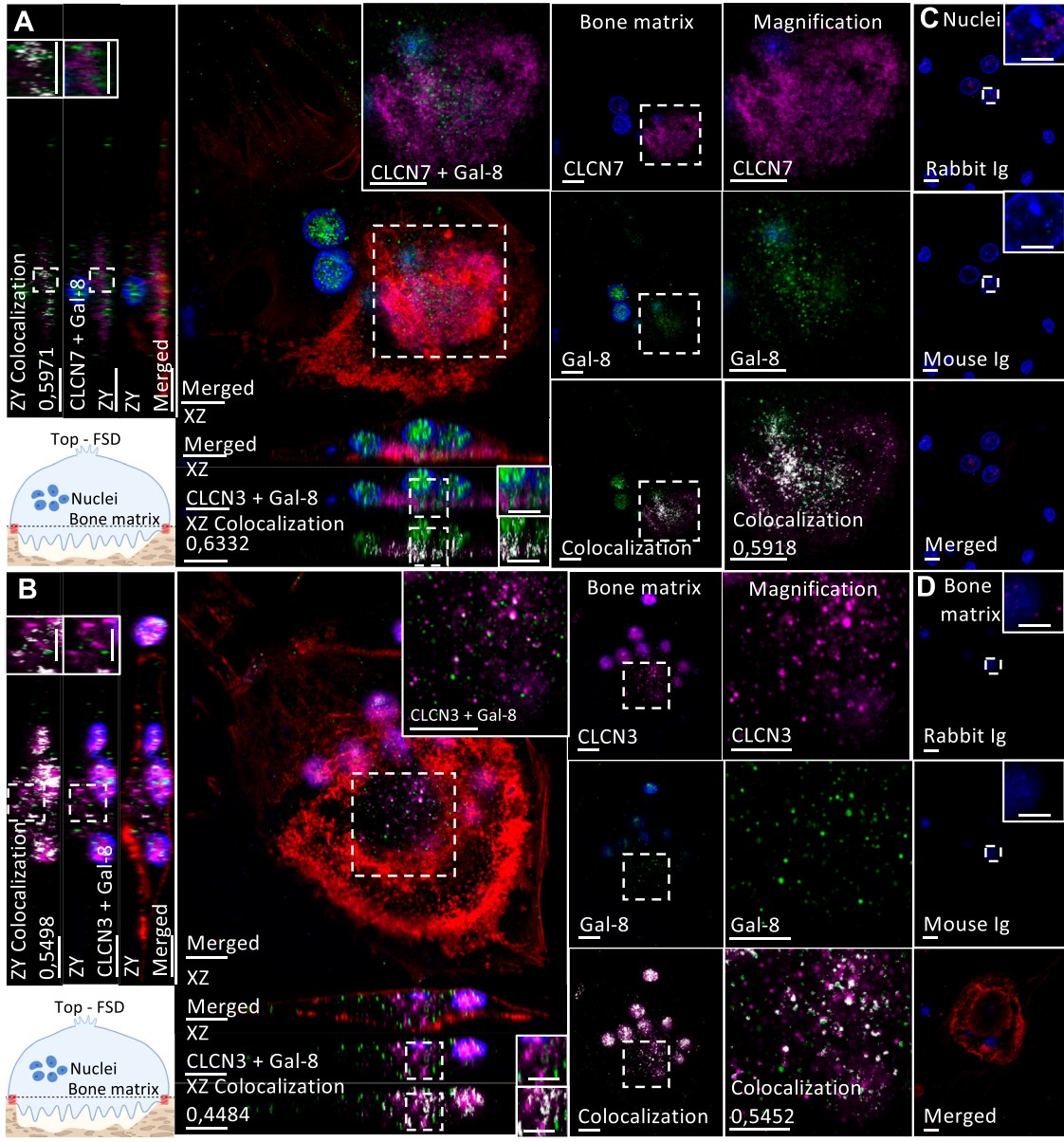

**Figure 8. Co-immunofluorescence staining of galectin-8 and the chloride channels CLCN3 and CLCN7.**
At day 21, human osteoclasts cultured on bone slices were double-stained using antibodies against galectin-8 (green) and CLCN3/CLCN7 (magenta), in addition to phalloidin Alexa Fluor 488 to detect F-actin (red) and delimit active osteoclasts and their sealing zones and DAPI (blue). The Mander overlap coefficient (R) is indicated when relevant. Magnified areas are delineated by dashed lines where applicable. **(A, B)** CLCN7 and CLCN3 distribution: optical sections at the bone matrix level are shown with orthogonal reconstruction of the presented cells and magnification the galectin-8 and CLCN7 (A) or CLCN3 (B) distribution at the ruffled border with their co-localization overlay (scale bar: 10 μM). Images are representative of three independent experiments with the acquisition of at least 10 osteoclasts per experiment. **(C, D)** Rabbit and mouse Ig controls are represented at the nuclei (C) and bone matrix levels (D), in the presence of phalloidin and DAPI to confirm the acquisition of an active osteoclast.

matrix and from orthogonal cell reconstruction (R = 0.41–0.71) (Fig S5C and D).

## Discussion

Osteoclasts exhibit high mobility, transitioning between migratory and bone-resorbing states (Ory et al, 2008). This process involves internal trafficking pathways, including the fusion of secretory lysosomes with the ruffled membrane, crucial for bone resorption (Coxon & Taylor, 2008). Although galectins, a family of adhesive lectins, are largely involved in the regulation of cell adhesion, spreading (cytoskeleton reorganization), and migration, their involvement in the different facets and functions of osteoclasts remains poorly understood.

In this study, we confirmed the diffuse dot-like cytoplasmic pattern of galectin-8 expression in human osteoclasts and the predominance of the short isoform. Inhibition of galectin-8 expression resulted in the inhibition of bone resorption and the

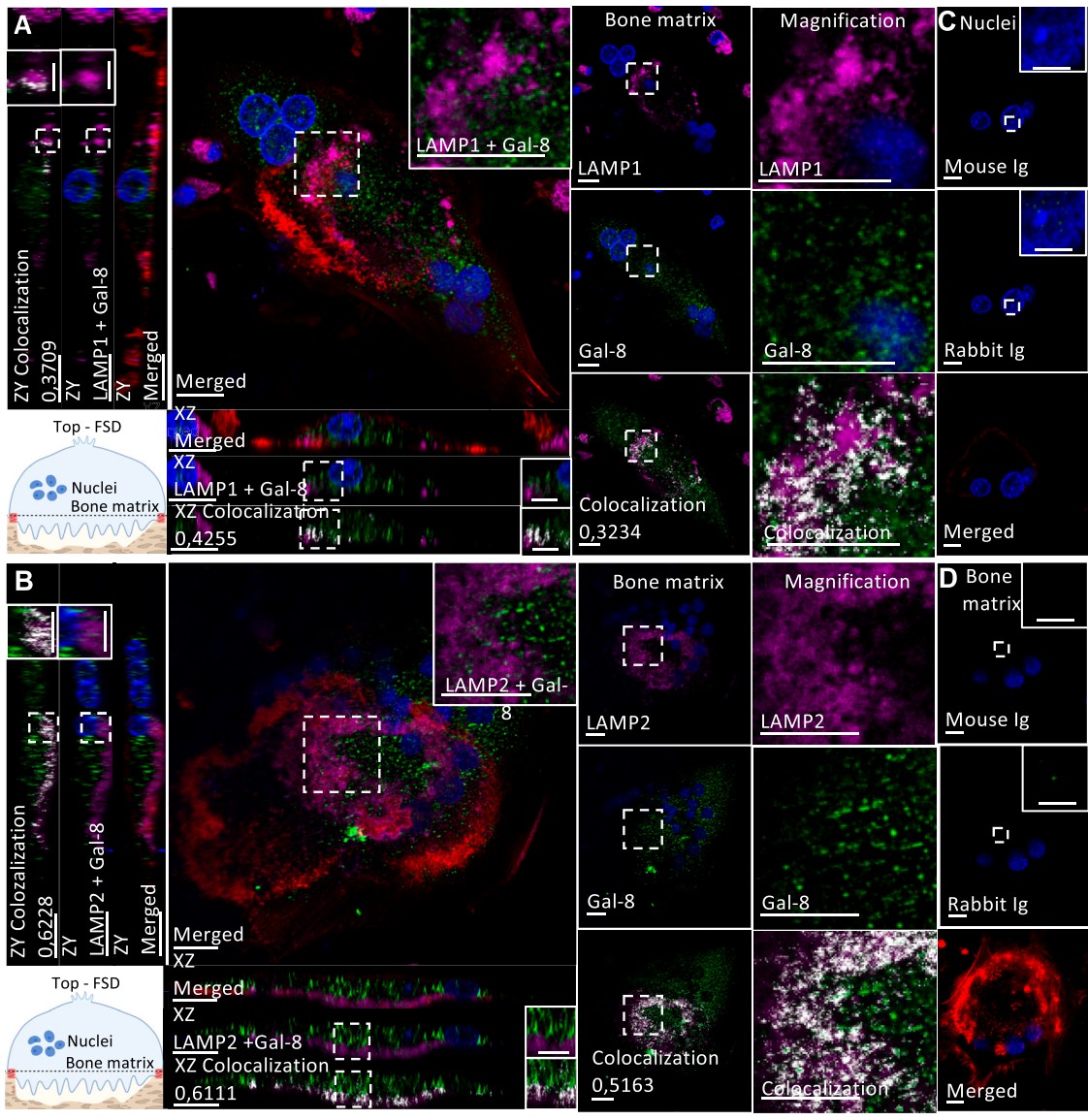

**Figure 9. Co-immunofluorescence staining of galectin-8 and the lysosomal membrane proteins LAMP1 and LAMP2.**
At day 21, human osteoclasts cultured on bone slices were double-stained using antibodies against galectin-8 (green) and LAMP1/LAMP2 (magenta), in addition to phalloidin Alexa Fluor 488 to detect F-actin (red) and DAPI (blue). The Mander overlap coefficient (R) is indicated when relevant. Magnified areas are delineated by dashed lines where applicable. **(A, B)** LAMP1 and LAMP2 distribution: optical sections at the bone matrix level are shown with orthogonal reconstruction of the presented cells and magnification the galectin-8 and LAMP1 (A) or LAMP2 (B) distribution at the ruffled border with their co-localization overlay (scale bar: 10 $\mu$M). Images are representative of three independent experiments with the acquisition of at least 10 osteoclasts per experiment. **(C, D)** Rabbit and mouse Ig controls are represented at the nuclei (C) and bone matrix levels (D), in the presence of phalloidin and DAPI.

stimulation of autophagy associated with inhibition of mTORC1 signaling and activation of AMPK. These two phenomena could be related as activation of autophagy inhibits bone resorption (Hussein et al, 2012). As ULK1 phosphorylation at Ser[757] by mTORC1 suppresses its catalytic activity, the low levels of pULK1[Ser757] upon galectin-8 knockdown reflect its impact on autophagy initiation. Our results contradict the reported intracellular role of galectin-8 as a stimulator of autophagy. Galectins classically inhibit mTORC1 and stimulate autophagy following the detection of lysosomal damage (Jia et al, 2018; Bell et al, 2021; Pied et al, 2022). Given the central role of autophagy in cellular homeostasis and the diverse

functions of galectins, the resulting effects of galectin-8 and other galectins depend on cell types and culture conditions, with an impact on autophagy upon cellular stress or endosomal damage (Johannes et al, 2018). Conversely, some autophagy-related proteins, including LC3B, Atg5, and Rab7, are involved in forming the sealing zone and ruffled membrane, potentially being used in bone resorption within active osteoclasts (DeSelm et al, 2011; Chung et al, 2012). In the absence of lysosomal stress, galectin-8 could therefore promote resorption while inhibiting autophagy in osteoclasts.

Our results revealed a direct role of intracellular galectin-8 in promoting bone resorption in human osteoclasts, whereas only an

indirect role was observed via RANKL production in a transgenic mouse model overexpressing galectin-8 (Vinik et al, 2015). Although the impact on osteoclast differentiation cannot be assessed through late-stage transfection, a decrease in the number of MNCs at this stage may be attributed to either excessive fission, reduced multinucleation, or apoptosis of osteoclasts, potential outcomes subsequent to the resorption activity (McDonald et al, 2021). The decline in bone resorption could be in part due to the reduced number of osteoclasts, but the resorption process itself was also affected, showing a decreased resorption per MNC, primarily influenced by the short isoform. In bone-resorbing osteoclasts, the sealing zone comprises densely packed actin-rich podosomes that delimit the ruffled border. As osteoclasts transition to a non-resorptive or migrating state, this zone shifts to a more relaxed podosome belt (Ory et al, 2008). Our findings demonstrate that a decrease in galectin-8 leads to a relaxation of the sealing zone. This impact on the zone's structural integrity might involve integrins like β1, a well-known partner of galectin-8 (Carcamo et al, 2006), playing a crucial role in osteoclast adhesion and bone resorption (Helfrich et al, 1996).

Galectin CRDs are highly conserved among species (Johannes et al, 2018), but AS giving rise to the long isoform with 42 additional amino acids is specific to humans. The proresorptive effect of galectin-8 is enhanced by a decrease in the long/short isoform ratio. Indeed, in pagetic osteoclasts, up-regulation of galectin-8 and a decrease in long isoform expression (ratio of long isoform to total [long + short] isoform RNA < 10%) were associated with increased bone resorption, increased mTOR signaling, and a defect in autophagy (Klinck et al, 2014). Splice switches might represent a biological mechanism for regulating osteoclast activation and alternation between bone migration and resorption states through a downstream impact on protein–protein interactions and signaling pathways, as indicated by our in vitro experiments of down-regulation of the short galectin-8 isoform, leading to an inhibition of bone resorption and decrease in the number of MNCs, whereas autophagy was not isoform-dependent.

To further understand the role of galectin-8 isoforms in osteoclasts, proteomic analysis of their interaction partners was conducted. Few interacting proteins have been identified to date, and the interactors varied by cell type in prior research. Galectin-8 binding partners were identified in murine macrophages during intracellular bacterial infection by IP-LC/MS, including autophagy-related proteins such as TAX1BP1 and ubiquitin, and lysosomal proteins and membrane trafficking proteins, including Rab7 and Rab14 (Bell et al, 2021). In endothelial cells, the ligands for galectin-8 included podoplanin, CD44, and CD166 of the immunoglobulin superfamily (Troncoso et al, 2014). Galectin-8 also binds several integrins, mainly β1 integrins depending on the cell type (Hadari et al, 2000; Carcamo et al, 2006; Nishi et al, 2006), or NDP52 in selective autophagy (Thurston et al, 2012).

Our study therefore represents the first proteomic assessment of interactions as a function of the galectin-8 isoform. Our galectin-8 interactome analysis detected 22 proteins common to both isoforms and 9 proteins specific for the short isoform. Despite the identical presence of CRDs in galectin-8 short and long isoform samples, the short isoform had more interacting proteins than the long isoform, but the latter is known to be unstable (Levy et al,

2006), with a linker region sensitive to thrombin (Nishi et al, 2005). Apart from protein cleavage, which was not present in our in vitro system, the difference in interacting proteins between galectin-8 isoforms could be explained by spatial interference of the long flexible loop with some binding sites or the spacing of CRDs altering the conformational sites involving their proximity. The long isoform could thus block the binding of certain interaction partners, thereby inhibiting certain functions such as resorption; in other words, splicing of LGALS8 appears to be an activation pathway for bone resorption.

To validate the interactions of short isoform–specific proteins with galectin-8 in human osteoclasts, we identified short isoform–specific pathways associated with secretory vesicles (CLCN7, LAMP1). Among common interactors, LAMP2 and CLCN3 also contribute to bone resorption, suggesting potential involvement of both isoforms. However, the long isoform, which binds glycans more slowly, might be less effective (Zhang et al, 2015). This dual involvement helps explain why, in experiments focusing on bone resorption, the impact of the short isoform appears less significant compared with the overall effect of total galectin-8. In immunofluorescence studies (CLCN3/7, LAMP1/2), we observed that galectin-8 mainly co-localized with LAMP2 and CLCN7 at the ruffled membrane of active osteoclasts. In addition, it showed co-localization with CLCN3 in a dot-like staining pattern between nuclei and the bone matrix. LAMP1 was only detected in a few osteoclasts. Given that osteoclasts alternate rapidly between resorbing and migrating states, LAMP1 might transiently express during a state that was not predominantly represented in our culture conditions. As suggested in other studies, intracellular galectin-8 in human osteoclasts may assist in vesicle transport to the ruffled membrane, possibly via microtubules, contributing to cell polarization during bone resorption (Lim et al, 2017; Lo et al, 2021).

The multifactorial origin of PDB includes p62 mutations, genetics, environmental elements, and viral factors affecting osteoclast behavior (Chamoux et al, 2009; Galson & Roodman, 2014; McManus et al, 2016). Galectin-8 undergoes AS, resulting in a shorter variant associated with the osteoclast phenotype in PDB (Klinck et al, 2014), which intensifies bone resorption. Galectin-8 functions across multiple levels, some of which are specific to the short isoform: interacting with CLCN3/7 to acidify the resorption area, associating with LAMP1/2 in secretory vesicles for ruffled membrane formation. In addition, galectin-8 may impact the sealing zone through β1 integrin interactions. Ultimately, the decrease in nuclei count per osteoclast because of lowered galectin-8 levels, especially its short isoform, suggests the potential influence of galectin-8 on the heightened multinucleation seen in Paget's disease osteoclasts. Given the diffuse localization of galectin-8 in human osteoclasts, it is likely that additional interacting partners remain to be identified, thereby helping to further delineate the functions of galectin-8 in osteoclast biology.

Although conventional methods present challenges in transfecting osteoclasts (Laitala-Leinonen, 2005), our transfection study faces a limitation because of the incomplete elimination of galectin-8 and its isoforms. This incompleteness might hinder the interpretation of each isoform's role. Nonetheless, noteworthy reductions were observed: a 75% decrease in total galectin-8, 73% in the long isoform, and 64% in the short isoform. These reductions were sufficient to induce significant functional changes, although

they led to only a 20–25% alteration in the isoform ratio. Nevertheless, small changes in spliced isoforms can induce significant functional changes, with AS events leading to fundamentally different functions or interactions (Schwerk & Schulze-Osthoff, 2005).

In conclusion, our study highlights the potential role of AS in regulating the functions of human osteoclasts and reveals galectin-8 as a new player in bone dynamics, contributing to PDB, and paving the way to investigate new mechanisms involved in the osteoclast phenotype.

# Materials and Methods

### Reagents

Opti-MEM was purchased from Gibco (Thermo Fisher Scientific). EMEM, antibiotics (penicillin, streptomycin, fungizone), and FBS were obtained from Wisent (Montreal, QC, Canada). Ficoll-Paque was purchased from Cytiva (Marlborough, MA, USA). MethoCult was acquired from StemCell Technologies. 12-well plates, eight-well chamber slides, and culture dishes were procured from Falcon (Corning). Modified Mayer's Hematoxylin (Lillie's Modification) was purchased from Dako (Agilent Technologies). Human recombinant (hr) M-CSF and hrGM-CSF were purchased from R&D Systems (Bio-Techne). Soluble hrRANKL was produced in-house as described elsewhere (Manolson et al, 2003). Detailed information on the antibodies used for immunoblotting and immunofluorescence is available in Tables S4 and S5, respectively. Lipofectamine LTX and Qubit Protein Broad Range Assay were purchased from Invitrogen (Thermo Fisher Scientific); anti-FLAG M2 magnetic beads were obtained from Sigma-Aldrich (Millipore Sigma); DsiRNA and custom siRNA were acquired from Integrated DNA Technologies; and negative-control siRNA was purchased from QIAGEN (sequences provided in Table S6).

### Osteoclast cultures

We used cord blood monocytes (CBMs) as osteoclast precursors. Umbilical cord blood was obtained from healthy parturient women and harvested at delivery after obtaining informed consent, as approved by our institution's review board (CRCHUS). CBMs were isolated from heparinized blood by density-gradient centrifugation and seeded on a methylcellulose-based medium for 10 d. CBMs were then collected and seeded in eight-well culture slides or 12-well plates for an additional 10 d in Opti-MEM supplemented with 2% FBS, M-CSF (25 ng/ml), hrRANKL (100 ng/ml), and antibiotics. In these long-term (21-d) primary cultures, mononuclear osteoclast precursor cells fuse, progressively adopting osteoclast markers, with bone resorption ability being their sole specific marker (Galson & Roodman, 2010; Roy et al, 2021). In all experiments, the number of independent experiments or biological replicates (the number of different blood samples used for culture) is specified, and all conditions are performed in duplicate (technical variability). We set out to study galectin-8 expression in osteoclasts, an area that remains largely unexplored. For this reason, protein expression analysis was initially conducted at various time points during the 21-d culture. However, for all subsequent experiments, such as knockdown procedures and immunofluorescence assays, our focus was specifically on mature osteoclasts.

### Immunoblotting

Whole-culture proteins were extracted from 12-well plates at the end of the 3-wk culture (day 21) or at different stages of differentiation within a single experiment (both at the outset and at 7, 14, or 21 d of culture), quantified using the Qubit Protein Assay and separated by SDS–PAGE. Immunoblotting was performed by incubating membranes with primary antibodies overnight at 4°C (Table S4). HRP-conjugated secondary antibodies were used for detection using a chemiluminescent system. Anti–$\beta$-actin antibody was used as a loading control. The optical density of the bands was quantified using NIH ImageJ software and normalized to that of the $\beta$-actin bands.

### Immunofluorescence and confocal microscopy

At the end of the maturation period, osteoclasts seeded on bone slices were quickly washed with PBS and fixed with 3% PFA. After permeabilization using 0.1% Triton X-100 and autofluorescence quenching using 0.25 M glycine, nonspecific binding sites were blocked using a ready-to-use Protein Block Serum Free solution (Dako). Specific antibodies directed against galectin-8, CLCN3, CLCN7, LAMP1, and LAMP2 (Table S5) were incubated in a ready-to-use antibody diluent with background component solution (Dako) overnight at 4°C. After washing, cells were incubated with Texas Red anti-mouse (orange), Alexa Fluor 647 anti-rabbit (near infrared), and phalloidin Alexa Fluor 488 (green). Cells were also counterstained with 0.5 ng/ml DAPI (Sigma-Aldrich) to visualize the nuclei. After 1 h of incubation, cells were washed several times with PBS and mounted using VECTASHIELD (Vector Laboratories). Cells were examined using a scanning confocal microscope (Leica TCS SP8, Leica Microsystems, Wetzlar, Germany) coupled to an inverted microscope with an HC PL APO ×63 oil immersion objective (Photonic microscopy platform, UdS, Leica Microsystems). Digitized images were computed and processed using Leica LASX software as previously described (Roy et al, 2021). In immunofluorescence double-labeling, an intermediate color indicating co-localization is obtained only if the fluorescence emission intensities of the two probes are similar, which is not the case for our antibodies. Therefore, we used white in the overlay picture for pixels with positive signals for both probes. Co-localization was quantified using Mander's overlap coefficient (R, range, 0–1) (Bolte & Cordelieres, 2006) computed from fluorescence imaging to determine the proportion of overlapping pixels between the two channels in specific region of interest of plan Z and orthogonal reconstruction (Leica LASX software). The integrity of the sealing zone was evaluated by measuring the area defined by the annular sealing zone of the maximum osteoclast projection, encompassing all Z plans using F-actin signal near the bone matrix (ImageJ). The average distance between podosomes was determined by measuring the distance between intensity peaks (podosomes) in a 10-$\mu$M horizontal section of the fluorescence intensity profile using Leica LASX software. For immunofluorescence studies, a minimum

of 10 osteoclasts per condition from three independent experiments were included.

### RNA interference

To investigate the roles of galectin-8, mature osteoclasts were either not transfected or transfected with a negative control DsiRNA, or with DsiRNA sequences down-regulating both isoforms of human *LGALS8* (Gal-8 DsiRNA). To optimize the inhibition of *LGALS8* expression, two sequences targeting different regions of the coding sequences of both isoforms were used. When studying the effects of each isoform, cells were either not transfected or transfected with negative control siRNA or siRNA sequences targeting specific regions of each isoform. The transfections occur in a late stage, near the end of osteoclast maturation (day 17), to examine how RNA interference affects both the function and phenotype of fully developed osteoclasts (day 21). All sequences are provided in Table S6. Transfection was performed according to the manufacturer's protocol; that is, the cell culture medium was replaced with serum and antibiotic-free Opti-MEM containing a mixture of 0.2 $\mu$M DsiRNA or siRNA and Lipofectamine LTX. After 4 h of incubation at 37°C, the transfection mixture was replaced with fresh medium, and cells were cultured for five more days. Protein down-regulation of galectin-8 and its isoforms was assessed by immunoblotting.

### Analysis of the number of MNCs

To calculate MNC numbers, cells were fixed using 3% PFA, permeabilized using 1% Triton X-100, stained using 10% modified Mayer's Hematoxylin followed by 0.25% eosin, and preserved in 50% glycerol mounting medium. Slides were visualized using a Zeiss ApoTome 2 microscope (Photonic microscopy platform, University of Sherbrooke, Sherbrooke, QC, Canada), and the total number of MNCs containing three or more nuclei was counted for each condition (area analyzed per well: 0.4 $cm^2$), along with the number of nuclei from 100 MNCs.

### Bone resorption

After extraction from MethoCult, CBMs were allowed to settle on devitalized bovine bone slices (0.3 $cm^2$ × 0.15 mm) and cultured for 2 wk as previously described (Roy et al, 2021). After removal from the cell culture, the bone slices were quickly washed in sodium hydroxide, sonicated in distilled water to remove cell debris, and stained with 1% toluidine blue solution containing 1% sodium borate. Dyed bone slices were observed under a light microscope with epi-illumination (Zeiss Semi 2000-c stereomicroscope, magnification ×45), and photos were taken. Resorption pits were defined by purple areas with a sparkling appearance and quantified using NIH ImageJ software.

### Autophagy and mTORC1 signaling

After osteoclast cultures were transfected with control DsiRNA or siRNA (Ctrl) or galectin-8 DsiRNA sequences (Gal-8) or isoform-specific siRNAs (Short, Long), proteins were extracted at (day 21), analyzed by immunoblotting using antibodies against the phosphorylated (p) or total forms of several proteins involved in mTORC1 signaling, including mTORC1 inhibitors AMPK$\alpha$ and T$^{172}$AMPK$\alpha$ and its target Raptor and Ser$^{792}$p-Raptor; mTORC1 targets: ULK1 and Ser$^{757}$p-ULK1, p70S6K and Thr$^{229}$p-p70S6K, 4E-BP1 and Thr$^{37}$/Thr$^{46}$p-4EBP1; upstream mTORC1 activators: PDK1 and Ser$^{241}$p-PDK1, Akt and Ser$^{473}$p-Akt, ERK and Thr$^{202}$/Tyr$^{204}$p-ERK. To study autophagy, we evaluated the expression of LC3B, a reliable marker associated with the development of autophagic structures, using immunoblotting. The autophagy flux was evaluated in conditions that promoted autophagy, using either starvation in serum-free EMEM or 10 $\mu$M rapamycin (a potent mTORC1 inhibitor, used as a positive control for autophagy), in the presence or absence of 40 $\mu$M chloroquine for 3 h, a lysosomotropic agent that inhibits lysosome-mediated proteolysis (Klionsky et al, 2021).

### HEK293T cell culture and transfection

Human embryonic kidney 293T (HEK293T) cells were cultured in DMEM supplemented with 10% FBS at 37°C in a humidified atmosphere containing 5% $CO_2$. Transient plasmid transfections were carried out using the *Trans*IT-LT1 reagent (Mirus Bio) according to the manufacturer's instructions. Empty pcDNA3 vector was added to each transfection preparation to ensure a constant quantity of DNA in each culture dish. Protein lysates were collected after 48 h.

### Plasmid constructs

The cDNA sequences encoding for the *LGALS8* short (NM_201543.2) and long (NM_006499.4) isoforms were amplified from a human leukocytes MATCHMAKER cDNA library (Clontech). The FLAG-tagged *LGALS8* constructs were generated by PCR using the Q5 high-fidelity DNA Polymerase (New England Biolabs) and primers containing the sequence encoding for the FLAG epitope in-frame with the N-terminal open-reading frame. The PCR fragments were digested with *Kpn*I and *Not*I and inserted into the pcDNA3 vector previously digested with the same restriction enzymes. The integrity of the coding sequences of all constructs was confirmed by sequencing at Génome Québec (McGill University, Montreal, QC, Canada) to confirm the integrity of the two constructs. Note that both generated *LGALS8* constructs presented the following single nucleotide polymorphisms (SNPs) classified as natural variants: dbSNP: rs1126407 (F19Y), dbSNP:rs1041935 (R36C), dbSNP:rs1041937 (M56V), and dbSNP:rs2243525 (R184S or R226S for the short and long isoform, respectively).

### Affinity purification–mass spectrometry (AP-MS)

HEK293T cells were transiently transfected with pcDNA3 (negative control), FLAG-*TBC1D25* (gifted by Steve Jean's laboratory, used as a control for the FLAG epitope), or FLAG-tagged *LGALS8* short (*LGALS8S*) or long (*LGALS8L*). Samples analyzed by LC-MS/MS were processed as previously described with some minor modifications (Degrandmaison et al, 2020; Frechette et al, 2021). At 48 h post-transfection, cells were washed in PBS and then incubated in solubilization buffer (1% octyl B-ᴅ-glucopyranoside, 75 mM Tris–HCl [pH 8], 2 mM EDTA, 5 mM MgCl$_2$) supplemented with protease

inhibitors (complete EDTA-free protease inhibitor cocktail supplemented with NaF, PMSF, $Na_3VO_4$, and $\beta$-glycerophosphate) at 4°C for 1 h. Supernatants were collected and quantified using the Qubit Protein Assay. Equal quantities of protein from each sample were incubated overnight with 75 $\mu l$ of anti-FLAG M2 magnetic beads at 4°C. Subsequently, the beads were washed four times in solubilization buffer without protease inhibitors and then four times with an ammonium bicarbonate buffer (20 mM $NH_4HCO_3$, pH 8.0). Immunoblotting was performed to validate the immunoprecipitation. Samples were then processed for liquid chromatography–mass spectrometry (LC-MS/MS) for the study of protein–protein interactions. The following steps were carried out by the Proteomics Platform (University of Sherbrooke, QC, Canada). Trypsin was added to the beads for a minimum of 5 h, and protein digestion was stopped by acidification with 1% formic acid solution. The supernatants were then transferred to LoBind tubes. The beads were incubated for 5 min in a solution of 60% acetonitrile and 0.1% formic acid, and the supernatants were collected again and pooled in LoBind tubes. The samples were dried, resuspended in sample buffer (0.1% trifluoroacetic acid), and desalted using a ZipTip. Finally, the samples were separated using HPLC Ultimate 3000 Binary RSLCnano (Thermo Fisher Scientific) and injected into a Q Exactive Hybrid Quadrupole-Orbitrap mass spectrometer (Thermo Fisher Scientific). Protein identification was performed using MaxQuant and the Human Proteins UniProt Knowledgebase at a false discovery rate of 1% (Leblanc & Brunet, 2020).

### Identification of HCIPs for galectin-8 isoforms

The selection of biological replicates taken in pairs was based on a variation coefficient lower than 20–25% for a Spearman rank correlation coefficient exceeding 0.75. Five replicates were used as negative controls (empty vector control or FLAG-TBC1D25 control), and two replicates were generated for each isoform. Protein–protein interaction candidates from AP-MS data were scored using the SAINTexpress algorithm (Teo et al, 2014). Confident interactors were selected based on a SAINT score of ≥0.9 (stringent SAINT score) with one of the two isoforms and >0.7 (threshold SAINT score) with the other isoform. The STRING database (https://string-db.org/) was used to detect interactions between proteins that interact with galectin-8 retaining a combined score greater than 0.75. The interaction network between the isoforms and their partners was constructed using Cytoscape version 3.9.1.

### Interaction network analysis using Gene Ontology (GO) and KEGG

We used the "clusterProfiler" R package for GO and KEGG pathway enrichment analyses of interacting proteins (Wu et al, 2021). To compute the enriched annotations, the parameter for the "enrichGO" and "enrichKEGG" functions was an adjusted *P*-value of the false discovery rate method with a Benjamini–Hochberg correction lower than 0.01. The "Heatmap" function of the "ComplexHeatmap" package was used for the graphical representation of the enriched terms as a function of $-\log_{10}$ (*P*-adjust).

### Statistical analysis

Results are presented as the mean ± SD. Significant variations were analyzed using paired one-way ANOVA followed by Tukey's test. Two-way ANOVA followed by Tukey's test was used to analyze significant variations for the splicing profile. The chi-squared test with Bonferroni's correction was used to assess significant variation in the MNC proportions of low-, medium-, or high-nucleated cells. Statistical significance was defined as $P < 0.05$. All statistical analyses were performed using GraphPad Prism software 9.0 (GraphPad, San Diego, CA, USA).

## Data Availability

The datasets and supplemental information can be acquired by contacting the corresponding author. All data identifying HCIPs and the complete results of enrichment analyses are available (Supplemental Datas 1 and 2).

## Supplementary Information

## Acknowledgements

S Roux was supported by the Clinician Scientist Fellowship Program of the Department of Medicine, Faculty of Medicine, Sherbrooke University, and the study was granted by the Natural Sciences and Engineering Research Council of Canada (NSERC) (#RGPIN-2016-03900). MA Brunet is a junior research scholar from the Fonds de recherche du Québec—Santé (FRQS). L Mbous Nguimbus received a scholarship of excellence from the Faculty of Medicine, and J Degrandmaison received a PhD scholarship from the FRQS.

### Author Contributions

M Roy: conceptualization, formal analysis, validation, investigation, visualization, methodology, and writing—original draft, review, and editing.
L Mbous Nguimbus: software, formal analysis, validation, investigation, visualization, methodology, and writing—review and editing.
PY Badiane: investigation, methodology, and writing—review and editing.
V Goguen-Couture: investigation, methodology, and writing—review and editing.
J Degrandmaison: formal analysis, investigation, methodology, and writing—review and editing.
J-L Parent: conceptualization, resources, formal analysis, supervision, and writing—review and editing.
MA Brunet: conceptualization, resources, data curation, software, formal analysis, supervision, validation, investigation, visualization, methodology, and writing—review and editing.
S Roux: conceptualization, data curation, formal analysis, supervision, funding acquisition, validation, investigation, methodology,

project administration, and writing—original draft, review, and editing.

## Conflict of Interest Statement

The authors declare that they have no conflict of interest.

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
