## [Reviewer comments · Life Science Alliance]

Life Science Alliance

Galectin-8 modulates human osteoclast activity partly through isoform-specific interactions

Michèle Roy, Léopold Mbous Nguimbus, Papa Yaya Badiane, Victor Goguen-Couture, Jade Degrandmaison, Jean-Luc Parent, Marie Brunet, and Sophie Roux

DOI: <https://doi.org/10.26508/lsa.202302348>

Corresponding author(s): *Sophie Roux, Université de Sherbrooke*

Review Timeline:

Submission Date:	2023-08-30
Editorial Decision:	2023-10-13
Revision Received:	2024-01-11
Editorial Decision:	2024-02-09
Revision Received:	2024-02-12
Accepted:	2024-02-13

Transaction Report:

October 13, 2023

Re: Life Science Alliance manuscript #LSA-2023-02348-T

Dr. Sophie Roux
Sherbrooke University
Rheumatology
3001, 12th avenue north
Sherbrooke, Quebec J1H5N4
Canada

Dear Dr. Roux,

Thank you for submitting your manuscript entitled "Galectin-8 modulates human osteoclast activity partly through isoform-specific interactions" to Life Science Alliance. The manuscript was assessed by expert reviewers, whose comments are appended to this letter. We invite you to submit a revised manuscript addressing the Reviewer comments.

Thank you for this interesting contribution to Life Science Alliance. We are looking forward to receiving your revised manuscript.

Sincerely,

B. MANUSCRIPT ORGANIZATION AND FORMATTING:

Reviewer #1 (Comments to the Authors (Required)):

Galectins are a group of proteins that have an evolutionary history and bind to β -galactosidase through carbohydrate-recognition domains. This family of proteins includes various members, each with specific roles. Previous research focused on galectin 3's role in bone and osteoclasts using galectin 3 null mice. This study, however, centers on galectin 8, a multifunctional protein involved in cell adhesion, inflammation, immune responses, and autophagy modulation. A previous study identified an alternatively spliced form of galectin 8, LGALS8, associated with pagetic osteoclasts. This work explores the expression patterns of long and short galectin 8 isoforms and their impact on human osteoclasts. LC-MS/MS techniques helped identify binding partners. Both long and short galectin 8 isoforms share many binding partners, but the short isoform exclusively associates with ITG family cell adhesion proteins and lysosomal protein LAMP1. Overall, this is thorough study and the premise of this study and its approach is sound, however the further characterization of osteoclasts would enhance the manuscript.

The study demonstrates that knockdown of gal-8 and/or its short isoform reduces osteoclast bone resorption when adjusted for multinucleated cell numbers. However, the data presented does not explain these defects adequately. Experiments monitoring F-actin rings to examine sealing zone, and acidification levels could be considered to support the conclusion that gal-8 modulates bone resorption activity.

Markers of osteoclast differentiation, such as DC-STAMP and NFATc1 expression, should be included to confirm cell differentiation and maturation status, considering the observed reduction in multinucleated cells/fusion following gal-8 depletion.

It's essential to acknowledge the limitations of the study, notably the modest reduction achieved through RNA interference to downregulate galectin isoforms, which raises questions about specificity and reliability. This limitation should be discussed.

The discussion section, could be improved to include the potential implications of these discoveries in the context of pagetic osteoclasts.

Reviewer #2 (Comments to the Authors (Required)):

Dear Editor and Researchers

The paper submitted by Roy and colleagues presents new knowledge of primary human osteoclast function (in vitro). An aspect being important for future research and future treatment of bone diseases. The authors present extensive and thorough results from in vitro work with primary human osteoclasts.

Major concerns:

I have some major concerns about the presentation of methods and data, which will be important to address thereby enabling transparency and reproducibility:

Methods:

The methods should be elaborated and clearer. Parts of most figure legends can easily be moved into the methods section, thereby clarifying the methods and simplifying the figure legends.

A major concern, however, is statement of number of experiments and replicates. In In Vitro studies like this, it is important to clearly state the number and use of:

- Biological replicates, i.e. from how many different donors were these experiments performed. If different numbers of donors were used in different experiments, this should be clearly stated.
- Experimental replicates, where experiments repeated using cells from the same donor - and in that case how many passages did the cells pass between experiments
- Technical replicates, only in one figure legend technical replicates are mentioned as "two bone slices per condition", which seems like a quite low number.

Without this knowledge it is hard to evaluate the true strength of this study. The authors use the phrase "n independent experiments", without elaborating what the replicates represent. This is not stated in the methods section either.

Methods should be elaborated, reproducible and more clear. On the other hand, figure legends are too complicated, because of too much method being included - thereby taking away the focus from the very nice results presented.

Minor comments:

Figures:

Fig1: It seems unnecessary to shown both 1C, D and E.

"based on three independent replicates" are these technical replicates or biological replicates? Western blot could be presented in supplementary figure, since the difference in splice variants are more interesting (just a suggestion for simplification).

Fig 2: Legend gets confusing because it shifts between having letters before and after respective description. Again, technical or biological replicates information is needed, either here or in the methods section.

Fig 4: Again placement of letters easily leads to confusion when interpreting the figures. Technical replicates or biological replicates? Two bone slices per condition??

Fig 5: Intuitive and consistent build up of figure legend - however a bit too much methodological information, which on the other hand is lacking in the methods section.

Fig 7: Again methodological heavy legend, which removes the focus from understanding the figures. Letters are again not placed intuitively when describing the figure - rather the method (e.g. 7A).

Results:

Section 1: "Using a specific antibody recognizing both isoforms" This sentence indicates that the figure shows where or how the antibodies recognizes each isoform, but this is not the case.

Figure 2A-D are supporting the method and could be supplementary figures.

Discussion:

I could be interesting to go more into detail about the differences between osteoclastogenesis and resorptive potential/activity. In the discussion this is only briefly mentioned. I would like some thoughts about the authors experimental setup with a quite long differentiation period (21 days).

Has Galectin-8 expression been correlated with other human bone diseases (e.g. from GWAS studies)?

Reviewer #3 (Comments to the Authors (Required)):

This is to investigate function of galectin-8, especially its short form splicing variant, on bone resorptive activity of human osteoclasts. The authors' group and others have found there are several alternative splicing variants in galectin-8. Here, they focused on two isoforms, long and short forms, respectively. Through gene silencing experiments specific for each isoform, the authors find that the short form may play a more robust role for osteoclast differentiation and subsequent bone resorbing activities. Through proteomics approaches, the authors find interacting proteins to galectin-8, which commonly or differentially interact with 2 isoforms. They further investigate 2 commonly and 2 short form-specifically interact proteins to demonstrate differences in their subcellular localization.

This is an interesting and unique finding. Exploring fine tuning mechanisms of osteoclastic bone resorption activity is important to understand that may lead to more efficient treatment regime for osteoporosis. Data shown in each figure well support the authors' interpretation. This reviewer has a couple of suggestions for clarity and readability.

1. Figure 2, it would be helpful to describe which region(s) of galectin mRNA is targeted to specifically silence each form.
2. Figure 3, please add meaning of # (probably "not significant") in the legend.
3. Figure 4, if both the short and long forms are silenced, is more robust reduction expected? Comparing the levels of reduction between panel B and D, the impact of silencing of the short form is far less than that of the total form, if the silencing efficiency is similar.
4. Figure 6, a cartoon to summarize the findings and the authors' conclusion would be helpful.
5. Figure 7, it is important to describe whether endogenous levels of galectin-8 in HEK293T cells are low enough to ensure the

specificity of interactive proteins from proteomics studies.

6. Figure 8/9, it is needed to mention in the figure legends what is the red color appeared in the merged photos in panel C/D.

Lastly, the authors should use "distribution" instead of "expression" when they describe localization of proteins.

Response to reviewers

We would like to thank the reviewers for their comments, which helped us to further improve the manuscript and its clarity (all the new changes made are shown in red throughout the new manuscript). Note that we have moved the results from the old Fig 2 to supplemental Fig S1, and we have added a new figure (Fig 4); consequently, the figure and supplemental figure numbers have changed.

Reviewer #1

1-The study demonstrates that knockdown of gal-8 and/or its short isoform reduces osteoclast bone resorption when adjusted for multinucleated cell numbers. However, the data presented does not explain these defects adequately. Experiments monitoring F-actin rings to examine sealing zone, and acidification levels could be considered to support the conclusion that gal-8 modulates bone resorption activity.

We agree that the impact on bone resorption is the result of the effect on osteoclasts and does not prejudice the mechanism involved. If the resorption adjusted to the number of osteoclasts is altered, the resorption process itself is affected. However, bone resorption comprises several major stages, which include adhesion and sealing zone formation, as well as acidification. In order to better understand the level of impact of galectin-8, we have expanded our work by analyzing its effect on the sealing zone. This does not exclude an impact at other sites, as suggested by the interactome.

Therefore, we investigated, via immunofluorescence, the sealing zone in human osteoclasts cultured on bone slices. We specifically studied the sealing zone during a decrease in galectin-8 or its specific isoforms.

We added a new Figure (Fig 4): Impact of galectin-8 inhibition on sealing zone integrity

Added to Results (page 7, top): “As a classical role of galectins is to promote adhesion, and in order to better understand how the resorption process could be affected by galectin 8, we investigated by immunofluorescence the sealing zone in osteoclasts cultured on bone (**Fig 4A**), measuring both the surface area of this zone (**Fig 4B**) and the density of the actin-rich podosomes comprising it (**Fig 4C**). Reducing galectin-8 expression resulted in a disorganized sealing zone, showing a decrease in area and an increased distance between podosomes. Targeting the expression of either the short or long isoform led to a similar decrease in the distance between podosomes.”

Added to the Discussion (page 14, line 6): “In bone-resorbing osteoclasts, the sealing zone comprises densely packed actin-rich podosomes that delimit the ruffled border. As osteoclasts transition to a non-resorptive or migrating state, this zone shifts to a more relaxed podosome belt (Ory et al., 2008). Our findings demonstrate that a decrease in galectin-8 leads to a relaxation of the sealing zone. This impact on the zone's structural integrity might involve integrins like $\beta 1$, a well-known partner of galectin-8 (Carcamo et al., 2006), playing a crucial role in osteoclast adhesion and bone resorption (Helfrich et al., 1996).”

2-Markers of osteoclast differentiation, such as DC-STAMP and NFATc1 expression, should be included to confirm cell differentiation and maturation status, considering the observed reduction in multinucleated cells/fusion following gal-8 depletion.

*We agree that the studied model was not clear. We have added a sentence in the materials and methods section to specify that our work focuses on mature osteoclasts, rather than osteoclastic differentiation.

Added in 'Osteoclast cultures' section page 18, line 14: “We set out to study Galectin-8 expression in osteoclasts, an area that remains largely unexplored. For this reason, protein expression analysis was initially conducted at various time points during the 21-day culture. However, for all subsequent

experiments, such as knockdown procedures and immunofluorescence assays, our focus was specifically on maturing osteoclasts.”

Added in ‘RNA interference’ section page 20, line 9: “The transfections occur in a late stage, near the end of osteoclast maturation (D17), to examine how RNA interference affect both the function and phenotype of fully developed osteoclasts (D21).”

As bone resorption function is the only specific marker of mature osteoclasts, to assess the impact of galectin-8 on these cells, bone resorption was our chosen evaluation. Therefore, we did not use other markers to assess the stage of differentiation.

*For a better presentation of the results, we have also added the sentence (page 6, line 7) : “To assess the impact of galectin-8 on the phenotype of mature osteoclasts, we evaluated multinucleation and bone resorption—the exclusive specific marker for mature osteoclasts—in cultures where osteoclasts were transfected during the late stages of their maturation”

*The comment regarding multinucleation is important. The number of multinucleated cells allows us to differentiate between a decrease in resorption activity versus a decrease in the number of osteoclastic cells (added to discussion page 14, line 6).

The impact on multinucleation was discussed; not only does the number of multinucleated cells decrease, but also the number of nuclei per multinucleated cell, and the processes of fusion/fission have been mentioned in the discussion (page 14, line 3): “While the impact on osteoclast differentiation cannot be assessed through late-stage transfection, a decrease in the number of multinucleated cells at this stage may be attributed to either excessive fission, reduced multinucleation, or apoptosis of osteoclasts, potential outcomes subsequent to the resorption activity”

However, the presented work does not involve a more in-depth analysis of these processes.

3-It's essential to acknowledge the limitations of the study, notably the modest reduction achieved through RNA interference to downregulate galectin isoforms, which raises questions about specificity and reliability. This limitation should be discussed.

We specified in the text that the decrease in isoforms in the presence of siRNA was incomplete, which could limit result interpretability. Nonetheless, the reduction in expression remains at 75% for total galectin-8, 73% for the long isoform, and 64% for the short isoform. The decrease in the ratio of isoforms is approximately 20-25%, limiting the interpretation of the impact of each individual isoform. This likely explains why the effects of the short isoform, while significant, are less pronounced than the total isoform, especially concerning bone resorption. Another explanation could be the probable role of both isoforms in bone resorption, at least in certain aspects, as common interacting proteins are involved in resorption.

Revised sentence, page 16-17 (last paragraph): “While conventional methods present challenges in transfecting osteoclasts (Laitala-Leinonen, 2005), our transfection study faces a limitation due to the incomplete elimination of galectin-8 and its isoforms. This incompleteness might hinder the interpretation of each isoform's role. Nonetheless, noteworthy reductions were observed: a 75% decrease in total galectin-8, 73% in the long isoform, and 64% in the short isoform. These reductions were sufficient to induce significant functional changes, although they led to only a 20-25% alteration in the isoform ratio.”

4-The discussion section, could be improved to include the potential implications of these discoveries in the context of pagetic osteoclasts.

We fully agree that an integrative link was missing; therefore, we added a paragraph at the end of Discussion to incorporate our findings into the profile of the Pagetic osteoclast:

Page 16, line 11: “The multifactorial origin of PDB includes p62 mutations, genetics, environmental elements, and viral factors affecting osteoclast behavior (Chamoux et al., 2009, Galson & Roodman, 2014, McManus et al., 2016). Galectin-8 undergoes AS, resulting in a shorter variant associated with the osteoclast phenotype in PDB (Klinck et al., 2014), which intensifies bone resorption. Galectin-8 functions across multiple levels, some of which are specific to the short isoform: interacting with CLCN3/7 to acidify the resorption area, associating with LAMP1/2 in secretory vesicles for ruffled membrane formation. Additionally, galectin-8 may impact the sealing zone through β 1 integrin interactions. Ultimately, the decrease in nuclei count per osteoclast due to lowered galectin-8 levels, especially its short isoform, suggests the potential influence of galectin-8 on the heightened multinucleation seen in Paget's disease osteoclasts.”

Reviewer #2

1-Major concerns: I have some major concerns about the presentation of methods and data, which will be important to address thereby enabling transparency and reproducibility:

Methods: The methods should be elaborated and clearer. Parts of most figure legends can easily be moved into the methods section, thereby clarifying the methods and simplifying the figure legends.

We acknowledge that certain aspects of our methodology made figure labeling complex. To streamline clarity, we relocated pertinent details from legends to the Materials and Methods section and omitted redundant descriptions.

Additionally, we've included comprehensive information about our model, specifically addressing the timing of transfection, which occurs post-differentiation. We've clarified our focus on studying the impact of galectin-8 (and its knockdown) on the phenotype of mature osteoclasts rather than on the differentiation process itself.

Added in Materials and Methods (page 20, line 9) : “The transfections occur in a late stage, near the end of osteoclast maturation (D17), to examine how RNA interference affect both the function and phenotype of fully developed osteoclasts (D21).”

What may also have been confusing is that initially we did study Galectin-8 expression during differentiation (at 0, 1, 2 and 3 weeks of culture), as this is a very novel description of this protein in osteoclasts. However, all subsequent work focuses on mature osteoclasts; we have added a sentence to clarify this point (page 18, line 9): « We set out to study Galectin-8 expression in osteoclasts, an area that remains largely unexplored. For this reason, protein expression analysis was initially conducted at various time points during the 21-day culture. However, for all subsequent experiments, such as knockdown procedures and immunofluorescence assays, our focus was specifically on maturing osteoclasts.”

2-A major concern, however, is statement of number of experiments and replicates. In In Vitro studies like this, it is important to clearly state the number and use of:

- Biological replicates, i.e. from how many different donors were these experiments performed. If different numbers of donors were used in different experiments, this should be clearly stated.***
- Experimental replicates, where experiments repeated using cells from the same donor - and in that case how many passages did the cells pass between experiments***

- Technical replicates, only in one figure legend technical replicates are mentioned as "two bone slices per condition", which seems like a quite low number.

Without this knowledge it is hard to evaluate the true strength of this study. The authors use the phrase "n independent experiments", without elaborating what the replicates represent. This is not stated in the methods section either.

Here too, we agree that the description of our samples lacked clarity. The number of independent experiments corresponds to the biological replicates, i.e. the number of different blood samples. For each blood, all experiments are performed in duplicate, which corresponds to the technical variability. On the other hand, we are working on primary cultures (which we have also clarified), using fresh umbilical cord blood, and it is not possible to have experimental replicates.

We have clarified this point by adding a sentence in Materials and Methods (page 18, line 11):

« We set out to study Galectin-8 expression in osteoclasts, an area that remains largely unexplored. For this reason, protein expression analysis was initially conducted at various time points during the 21-day culture. However, for all subsequent experiments, such as knockdown procedures and immunofluorescence assays, our focus was specifically on maturing osteoclasts.»

3-Methods should be elaborated, reproducible and more clear. On the other hand, figure legends are too complicated, because of too much method being included - thereby taking away the focus from the very nice results presented. (agree, see Rev2-comment 1)

4-Minor comments: Figures:

Many legends have been condensed by relocating method-related information to its respective section, and to prevent redundancy with the information already provided in the materials and methods section. The letters in Figures 1 to 9 were sometimes at the end, and we have rectified this.

Fig1: It seems unnecessary to shown both 1C, D and E.

"based on three independent replicates" are these technical replicates or biological replicates?

Western blot could be presented in supplementary figure, since the difference in splice variants are more interesting (just a suggestion for simplification). Fig 2: Legend gets confusing because it shifts between having letters before and after respective description. Again, technical or biological replicates information is needed, either here or in the methods section. Fig 4: Again placement of letters easily leads to confusion when interpreting the figures. Technical replicates or biological replicates? Two bone slices per condition??

We agree, Figure 1-D has been removed.

We have clarified the point concerning replicates by adding a sentence in Materials and Methods (Rev2- comment 2)

Fig 5: Intuitive and consistent build up of figure legend - however a bit too much methodological information, which on the other hand is lacking in the methods section

We agree, we've simplified the legend and, we've introduced a new section entitled "Autophagy and mTORC1 Signaling" in the "Materials and Methods" section.

Fig 7: Again methodological heavy legend, which removes the focus from understanding the figures. Letters are again not placed intuitively when describing the figure - rather the method (e.g. 7A).

Again, some parts were redundant, and we also added a sentence to the AP-MS section of the materials and methods: "Samples were then processed for liquid chromatography-mass spectrometry (LC-MS/MS) for the study of protein-protein interactions."

5-Results: Section 1: "Using a specific antibody recognizing both isoforms" This sentence indicates that the figure shows where or how the antibodies recognizes each isoform, but this is not the case. Figure 2A-D are supporting the method and could be supplementary figures.

We agree that the reference to (new) Fig. 1A was misplaced, a new sentence refers to the linear schematization of the 2 isoforms was added (page 5, line 4): «In Fig 1A, a linear representation showcases the two LGALS8 isoforms.»

We also have included the data from Figure 2A-D in a supplementary Figure (Fig S1) .

6-Discussion: I could be interesting to go more into detail about the differences between osteoclastogenesis and resorptive potential/activity. In the discussion this is only briefly mentioned. I would like some thoughts about the authors experimental setup with a quite long differentiation period (21 days)- Has Galectin-8 expression been correlated with other human bone diseases (e.g. from GWAS studies)?

Our model has been further detailed, particularly emphasizing our focus on mature osteoclasts, as well as highlighting the importance of timing for transfection (see Rev2-comment1). Osteoclastic differentiation in human models is more prolonged compared to mice; typically, a 3-week primary culture period is necessary to obtain fully differentiated osteoclasts.

Using the GWAS Catalog database or literature, we did not find any associations with other human pathologies."

Reviewer #3:

1. Figure 2, it would be helpful to describe which region(s) of galectin mRNA is targeted to specifically silence each form.

The precise recognition sites of the RNA interference are detailed in Supplemental Table 6. Additionally, the targeted regions of the *LGALS8* CDS for DsiRNA #1 and DsiRNA #2, as well as the mRNA targeted areas for each isoform-specific siRNA, have been highlighted in Figure 1 (shown as red lines) (Fig 1-A).

2. Figure 3, please add meaning of # (probably "not significant") in the legend.

We had omitted to specify the meaning of the symbol added to the legend of figure 3 (done).

3. Figure 4, if both the short and long forms are silenced, is more robust reduction expected? Comparing the levels of reduction between panel B and D, the impact of silencing of the short form is far less than that of the total form, if the silencing efficiency is similar.

We agree that if all resorption activity were linked to the short isoform, the decrease observed during the knockdown of the short isoform should equal that of the total galectin-8 knockdown. However, the reduction in resorption by MNC is lower (-26% vs -63%).

One reason is the incomplete specific decrease in the short isoform, approximately 64% compared to the 75% reduction in the total form. Moreover, the results from isoform-specific interactomes suggest potential involvement of both isoforms. We've observed a range of interaction partners, including proteins specifically associated with the short isoform like *CLCN7*, a key factor in bone resorption, along with other resorption-related proteins that bind to both isoforms (such as *LAMP2* and *CLCN3*). Consequently, our conclusion indicates that while galectin-8's impact on bone resorption is 'mainly' associated with the short isoform, it's not exclusively tied to it. Additionally, we've explicitly noted in the text that the long isoform might also play a role in this process.

We have clarified this "non-exclusive" aspect of the short isoform in the discussion, and added the following sentence (page 16): "This dual involvement helps explain why, in experiments focusing on bone resorption, the impact of the short isoform appears less significant compared to the overall effect of total galectin-8."

4. Figure 6, a cartoon to summarize the findings and the authors' conclusion would be helpful.

We have added a diagram to summarize the various proteins studied (Figure 5-A):

Page 7, line 7 from the bottom : "The different proteins related to mTORC1 that were studied have been represented on a diagram (Fig 5A)."

Legend Fig 5A: "(F) Diagram. The organizational chart of the mTORC1-related proteins /p-proteins studied is schematized."

5. Figure 7, it is important to describe whether endogenous levels of galectin-8 in HEK293T cells are low enough to ensure the specificity of interactive proteins from proteomics studies.

While HEK293T cells have been shown to express endogenous galectin-8 (Klinck et al., 2014), its expression appears undetectable when assessed alongside the high endogenous expression observed in pancreatic and lung carcinoma cell lines (Meinohl et al., 2019). However, for the interactome studies, constructs were tagged with FLAG, and the affinity purification of galectin-8 and its interacting partners was conducted using magnetic beads crosslinked to an antibody specific for the FLAG epitope. Consequently, if there is a low level of endogenous galectin-8 in HEK293T cells, it is absent in the samples processed for mass spectrometry and interactome analysis.

6. Figure 8/9, it is needed to mention in the figure legends what is the red color appeared in the merged photos in panel C/D.

Indeed, we had not specified that controls for primary and secondary antibodies (rabbit Ig and mouse Ig) were conducted in the presence of phalloidin, marking actin primarily at the level of the bone matrix sections (Figure 8-D and Figure 9-D), where the actin ring is located. We have added this clarification to the legends of Figures 8 and 9.

We added at the end of the figure 8/9 legends: .. "In presence of phalloidin and DAPI to confirm the acquisition of an active osteoclast."

7. Lastly, the authors should use "distribution" instead of "expression" when they describe localization of proteins. The term "distribution" is more appropriate than "expression" to describe IF, and has been included in the legend to figures 8 and 9.

February 9, 2024

RE: Life Science Alliance Manuscript #LSA-2023-02348-TR

Dr. Sophie Roux
Université de Sherbrooke
Rheumatology
3001, 12th avenue north
Sherbrooke, Quebec J1H5N4
Canada

Dear Dr. Roux,

Thank you for submitting your revised manuscript entitled "Galectin-8 modulates human osteoclast activity partly through isoform-specific interactions". We would be happy to publish your paper in Life Science Alliance pending final revisions necessary to meet our formatting guidelines.

- please address Reviewer 2's remaining comments
- please be sure that the authorship listing and order is correct
- please add the Twitter handle of your host institute/organization as well as your own or/and one of the authors in our system
- please use the [10 author names et al.] format in your references (i.e., limit the author names to the first 10)
- please remove legends from Supplementary figures; they should be provided only in the manuscript file

Figure Checks:

- in Figs 8B and 9D, the zoomed in box for Mouse Ig is completely black

A. FINAL FILES:

B. MANUSCRIPT ORGANIZATION AND FORMATTING:

Sincerely,

Reviewer #2 (Comments to the Authors (Required)):

Thanks for sending this reviewed manuscript - which has been significantly improved.

I have only minor comments:

P 5 l. 100: "using a specific antibody targeting both isoforms" but the figure shows isoform-specific results.

p 19 l. 430: "on maturing osteoclasts"

at what day of differentiation are the osteoclasts considered mature, and are they considered fully mature or still maturing?

Reviewer #3 (Comments to the Authors (Required)):

The authors made substantial changes according to the suggestions from this reviewer and others. The conclusion is further well supported by the additional data.

Response to reviewers

We would like to sincerely thank the reviewers for their comments, which allowed us to further improve the manuscript (all changes made are indicated in red in the new manuscript).

Editorial comments:

Fig 8B and 9D: the zoomed areas were improved because they were not on the right plane; this has been corrected (8C, 8D, and 9C). However, Figure 9D of mouse Ig remains indeed black because the zoomed area corresponds to non-specific signal for Rabbit IgG, and it is not located near a nucleus.

We have added a small sentence to explain the selection in the legend of these two figures: "Magnified areas are delimited by dashed lines where applicable."

Reviewer #2

P 5 l. 100: "using a specific antibody targeting both isoforms" but the figure shows isoform-specific results.

The anti-galectin 8 antibody used allows the recognition of both short and long isoforms differentiated by their molecular weight on the immunoblot (Fig 1B). We agree that the adjective 'specific' was misleading.

We have modified the sentence:

« Using a non-isoform-specific antibody, we therefore assessed galectin-8 and distinguished its long and short isoforms based on their molecular weights by immunoblotting at different time points during osteoclast long-term cultures »

**p 19 l. 430: "on maturing osteoclasts"
at what day of differentiation are the osteoclasts considered mature, and are they considered fully mature or still maturing?**

We agree that, once again, the terminology was prone to confusion. We wanted to emphasize that the study of osteoclasts is conducted starting from day 17 for transfections and day 21 for result analysis, at which point the majority of osteoclasts are considered mature.

During the first week, osteoclastic precursors proliferate and commit to the osteoclastic lineage. By the second week, osteoclastic differentiation continues, and committed cells begin to express osteoclastic markers while gradually losing macrophagic markers. Fusion and progressive multinucleation occur, and some bone resorption may already be observed in culture, increasing over time and peaking in the third week (*Hodge et al, JBMR 2004 # 14969388; Takahashi et al, Dev Biol 1994 # 8174777*).

Thus, during the final days of the culture period studied in our work, osteoclasts can be considered differentiated, with terminal maturation corresponding to the acquisition of bone resorption function.

Page 18: We have replaced "maturing" with "mature."

February 13, 2024

RE: Life Science Alliance Manuscript #LSA-2023-02348-TRR

Dr. Sophie Roux
Universita de Sherbrooke
Rheumatology
3001, 12th avenue north
Sherbrooke, Quebec J1H5N4
Canada

Dear Dr. Roux,

Thank you for submitting your Research Article entitled "Galectin-8 modulates human osteoclast activity partly through isoform-specific interactions". It is a pleasure to let you know that your manuscript is now accepted for publication in Life Science Alliance. Congratulations on this interesting work.

DISTRIBUTION OF MATERIALS:

Again, congratulations on a very nice paper. I hope you found the review process to be constructive and are pleased with how the manuscript was handled editorially. We look forward to future exciting submissions from your lab.

Sincerely,
